# Predicting rates of cognitive and functional decline in Alzheimer's disease and mild cognitive impairment

Antigone Fogel [1,2] ✉, Chloe Walsh [1,2,3], Nan Fletcher-Lloyd [1,2], Paresh Malhotra [1,2,4], Mina Ryten [5,6,7], Ramin Nilforooshan [1,2,3] & Payam Barnaghi [1,2] ✉

## Abstract

**Background** The global population of People Living with Dementia (PLWD) is expected to grow rapidly in the coming decades, increasing the need for personalised, generalisable, and scalable prognosis and care planning support. However, current prognostic guidance does not adequately capture the heterogeneity in dementia trajectories, and existing predictive models of dementia progression rely on costly and inaccessible data, limiting their scalability in resource-constrained settings.
**Methods** Using clinical assessments, demographic, and medical history data from 153 12-month clinical trajectories collected over three years, two machine learning algorithms were developed to predict 12-month cognitive and functional decline in Alzheimer's Disease (AD) and Mild Cognitive Impairment (MCI). Models were externally validated on 741 trajectories from the ADNI cohort. Cognitive and functional decline were estimated using the Mini-Mental State Exam (MMSE) and Bristol Activities of Daily Living (BADL).
**Results** The MMSE model achieves a mean absolute error (MAE) of 1.84 (95% CI: 1.64-2.04) internally and 2.19 in external validation. The BADL model achieves an MAE of 3.88 (95% CI: 3.46-4.30). Baseline scores on ideational praxis, orientation, and word recall are among the strongest predictors of cognitive decline, while independence in food preparation, finances, and dressing are among the top predictors of functional decline.
**Conclusions** Our models use only routinely collected and easily accessible data, offering high translational potential. If implemented, our scalable, data-driven prognostic support tool could streamline clinical workflows, support personalised care planning, and provide PLWD and their families with greater clarity and reassurance.

## Plain language summary

Nearly 60 million people are living with dementia right now. Dementia affects everyone differently, some people decline quickly, while others stay stable for longer. This makes it hard to plan suitable care for individuals and help them prepare for the future. We have followed a group of people living with dementia for over three years, regularly measuring their cognitive function and independence. Using this information, we built two algorithms that learn from their data to make personal predictions of how dementia will progress for others. To make our models easy for doctors to use in practice, we created an online app designed to help clinicians provide more personal guidance and support to their patients.

Trajectories of cognitive and functional decline vary considerably in Alzheimer's Disease (AD) and Mild Cognitive Impairment (MCI). However, this heterogeneity is not effectively accounted for in dementia care planning pathways[1]. Current prognostic methods rely on clinician experience and average rates of progression across people living with dementia (PLWD) with few opportunities for personalisation and no data-driven support[2,3]. In the absence of more tailored approaches, clinicians face challenges in planning appropriate care for their patients while PLWD and their families are

unable to make informed decisions about future financial and care-related concerns.

Currently estimated to affect nearly 60 million people worldwide, dementia is expected to have more than doubled in prevalence by 2050[4]. As the global population of PLWD rises, the need for personalised and scalable patient care will also rise. However, current estimates suggest that healthcare systems around the world are not sufficiently equipped to support this growing population [5,6]. Dementia care is complex and costly, often involving a coordinated effort from family members, clinicians, and professional

[1]Department of Brain Sciences, Imperial College London, London, UK. [2]UK Dementia Research Institute, Care Research and Technology Centre, London, UK. [3]Surrey and Borders Partnership NHS Foundation Trust, Leatherhead, UK. [4]Imperial College Healthcare NHS Trust, London, UK. [5]Department of Clinical Neurosciences, University of Cambridge, Cambridge, UK. [6]Department of Genomic Medicine, University of Cambridge, Cambridge, UK. [7]UK Dementia Research Institute, Cambridge, UK. ✉e-mail: antigone.fogel23@imperial.ac.uk; p.barnaghi@imperial.ac.uk

caregivers, and consideration of a person's unique combination of cognitive, functional, psychological, and physical needs[7]. Essential to planning appropriate care is an understanding of how each patient's condition will progress. However, with current prognostic methods, population-level averages fail to account for individual variability and the experience required to accurately predict personalised patient outcomes takes years to develop[8]. Thus, there is a need for accessible and tailored enhancements to existing care planning and provisioning approaches. Machine learning (ML) models trained on data from a large number of patient encounters and designed to reliably forecast clinical progression in dementia will be critical to addressing this need. Such methods can not only model patient-specific disease trajectories but also uncover particularly predictive clinical characteristics and complex interactions between them, all with improved efficiency, scalability, and cost-effectiveness[9].

Several Machine learning (ML) models have recently been developed with the purpose of predicting cognitive decline in dementia using a combination of clinical assessment, neuroimaging, and Cerebrospinal fluid (CSF) data[10–15]. These models have promising applications for clinical trial design and optimisation and have proven utility in modelling disease trajectories, but their application in clinical settings is limited. Neuroimaging and CSF biomarkers, although regarded as the gold standard for diagnosing dementia, are neither routinely collected nor do they provide cost-effective or scalable measurements of brain health[2]. In the majority of clinical settings, especially those far from high-resource urban centres, dementia is diagnosed and monitored using neuropsychological assessments and family and medical history information[1,16]. Thus, a majority of clinicians and their patients, by extension, would not be able to benefit from existing Machine learning (ML) models for disease course prediction, prognosis, or care planning.

Clinical assessment data have previously been used to predict individual risk of conversion from MCI to dementia[17], and age at onset of AD[18]. However, this data has not yet been used directly to predict changes in cognitive function or daily independence, two essential areas of focus during dementia care planning. In the present study, routinely collected clinical assessment data, specifically the sub-item level data that measures impairment in certain domains of cognition and function, are used to predict rate of change in cognitive and functional assessments at the individual level. This allows for personalised insights regarding disease progression to be generated, and specific domains of cognitive or functional impairment to be emphasised.

Recent work has also assessed the association between Electronic Health Record (EHR)-derived diagnosis and medication data and rate of cognitive decline, as measured by clinical assessments collected in memory clinic settings[19]. Similar to our approach, this study highlights the value of using existing clinical data to derive insights into the factors contributing to heterogeneity in dementia progression trajectories[19]. While Adams et al.[19] focus on population-level associations derived from EHR sources, our study aims to generate personalised predictions of disease progression by modelling baseline sub-domains of cognitive function and Activities of Daily Living (ADL).

Clinically applicable Machine learning (ML) models of decline in PLWD require several distinct considerations to make them usable in real-world and especially low-resource settings. First, model inputs must include only routinely collected and accessible data for clinicians in diverse environments[20]. Second, model decision-making should be explainable and interpretable to enhance clinical care without compromising safety, and finally, models should be easy for a clinician to use[20]. Models that take these considerations into account have demonstrated significant value in other contexts, not only by enhancing clinical care planning but also by empowering patients and their families with the knowledge to prepare for critical life transitions and providing a sense of control over their futures[21,22]. In this study, our aim was to develop clinically applicable Machine learning (ML) models to assess and estimate each patient's particular rate of decline in cognitive and functional abilities using easy-to-administer, routinely collected, and clinically validated neuropsychological assessments, medical history, and demographic data. We assessed performance of several traditional and explainable machine learning models and developed a clinical decision-support tool to make our trained prediction models interpretable and user-friendly for clinicians in various settings.

Cognitive decline is a defining feature of dementia progression, but impairments in other domains are also characteristic of the disease. As dementia progresses, a person's ability to perform fundamental ADL declines rapidly[23]. Monitoring and predicting both cognitive and functional decline is essential, as these changes inform care planning, determine the type of support needed, and directly influence independence and quality of life[7]. For this reason, two predictive models were developed to be used in tandem. One that predicts cognitive decline and another that predicts decline in ADL. Cognitive and functional impairment can be assessed in a variety of ways using more and less complex clinical tasks and assessments. In this study, models of cognitive and functional decline were trained to predict Mini-Mental State Examination (MMSE)[24] and Bristol Activities of Daily Living (BADL)[25] scores, respectively, 12 months in the future. Both assessments are easy, relatively quick to administer, and commonly used in clinical practice. Of the cognitive assessments in use across clinical settings, the MMSE and Alzheimer's Disease Assessment Scale–Cognitive Subscale (ADAS-Cog) are both collected for the cohort of participants in this study. Both are clinically validated and well-known assessments of cognitive function that provide distinct benefits and disadvantages[16]. While the ADAS-Cog is more granular than the MMSE, assessing a wider number of domains more precisely, it takes approximately five times longer to collect and is more commonly used in research settings, rarely in routine clinical practice[26]. The MMSE is a much shorter assessment, regularly collected to screen for dementia and to monitor changes over time. Because of its short nature, however, it lacks comparable granularity and sensitivity to the ADAS-Cog. Together, the two cognitive assessments provide complementary measurements of a person's cognitive function. While both assessments were included as model inputs, the MMSE was selected as the target variable for prediction because it is more widely used in clinical practice and provides clearer guidelines for stratifying severity of cognitive impairment[3]. While the utility of the MMSE, ADAS-Cog, and BADL in monitoring dementia progression are well documented[16,26,27], their relative value as predictors of future cognitive and functional abilities has not been explored as extensively. Beyond developing a strong predictive model of dementia progression, this study aimed to assess the predictive strength of each assessment and its subcomponents.

Clinical Decision Support Tools (CDST) play an important role in bridging the gap between academic discovery and clinical implementation[2,20], but while CDSTs are already in use in different areas of healthcare[28], no such tools are available for dementia care and clinical support. Here, we develop two clinically applicable ML models for dementia trajectory mapping that are designed for easy implementation into a wide variety of clinical settings. Their use will enhance existing dementia care planning pathways while supporting and empowering PLWD and their families.

## Methods
### Study design and cohort selection
Clinical assessment and medical history data used for model training were collected as part of the Minder Health Management Study (IRAS: 257561), an ongoing longitudinal follow-up study of PLWD in the United Kingdom, which began in 2018. Study participants include individuals with a diagnosis of dementia, MCI, frailty, stroke, or traumatic brain injury. Those with severe psychiatric symptoms, severe sensory impairment, or receiving treatment for terminal illness were excluded from the study. Participants were recruited from primary care, adult social care services, and memory clinics across Surrey and Borders Partnership National Health Service

## Table 1 | Clinical assessment overview and scoring

| Assessment & Domains assessed | Time Required | Score range | |
|---|---|---|---|
| MMSE | Orientation, registration, attention, recall, language, visuospatial impairment | 5–10 mins | 0–30 |
| ADAS-Cog 11 | Word recall, object naming, commands, ideational and constructional praxis, orientation, word recognition, spoken language, language comprehension, word finding, and remembering instructions | 45 min | 0–70 |
| BADL | Food and drink preparation, eating, drinking, dressing, hygiene, oral hygiene, bathing/showering, using the toilet, transfers, mobility, orientation (time & space), communication, telephone use, housework/gardening, shopping, managing finances, games/hobbies, and transport | 20 min | 0–60 |

*MMSE* Mini-Mental State Examination, *ADAS-Cog* Alzheimer's Disease Assessment Scale-Cognitive Subscale, *BADL* Bristol Activities of Daily Living.

(NHS) Foundation Trust and Hammersmith & Fulham Health and Care Partnership, North West London.

Diagnoses of dementia and other conditions were obtained through participants' electronic health records. In select cases, clinical dementia diagnoses were reassessed by the Minder research team via a Magnetic Resonance Imaging (MRI) scan and cognitive testing. If the research diagnosis did not match a participant's existing clinical diagnosis, the research diagnosis was used (MMSE model: $n = 13$, BADL model: n=10). Further details on how diagnoses were made can be found in the Supplementary Methods. For the purposes of this study, only Minder participants with a confirmed diagnosis of AD or MCI and more than 12 months of longitudinal data (before 25/03/2025) were included. A description of Minder participants who did not meet these criteria can be found in Supplementary table S3.

Data used for external model validation were obtained from the Alzheimer's Disease Neuroimaging Initiative (ADNI) database (adni.loni.usc. edu)[29]. The ADNI was launched in 2003 as a public-private partnership, led by Principal Investigator Michael W. Weiner, MD. The primary goal of ADNI has been to test whether serial magnetic resonance imaging (MRI), positron emission tomography (PET), other biological markers, and clinical and neuropsychological assessment can be combined to measure the progression of MCI and early AD. For the Minder and ADNI cohorts, only participants with all baseline assessments available and 12-months of follow-up data were included. A demographic overview of the model training and validation cohorts can be found in Table 2.

### Ethical approval
The Minder Health Management study received ethical approval from the London-Surrey Borders Research Ethics Committee; TIHM 1.5 REC: 19/LO/0102. The study is registered with the National Institute for Health and Care Research (NIHR) in the United Kingdom under the Integrated Research Application System (IRAS) registration number 257561. Upon joining the study, the participants and their study partners provided their written informed consent to participate in the study and for their data to contribute to publications. The ADNI study was approved by the Institutional Review Boards of all participating sites, all participants provided informed written consent.

### Data collection and pre-processing
**Demographic information.** All Minder participants completed a dementia proforma at baseline, which provided information about their age, sex, and other socio-demographic factors. Additionally, an age-sex interaction feature was created by first centring the binary sex feature such that females were assigned a value of −1 and males a value of 1. This numerical sex feature was then multiplied with a participant's age, resulting in an Age-sex interaction term where the lowest negative values represent the oldest women and the highest positive values represent the oldest men.

**Clinical assessments.** Regular clinical assessments were conducted to evaluate disease progression over time. Of these assessments, data from three that were directly related to cognitive and functional abilities were

used: The MMSE[24], ADAS-Cog[30,31], and BADL[25] questionnaires. Repeat MMSE and ADAS-Cog, but not BADL assessments, are collected regularly for ADNI participants. This allowed for validation of the model of cognitive but not functional decline developed in this study.

The 14-item ADAS-Cog is collected throughout the Minder study, but the 13-item ADAS-Cog is collected in Alzheimer's Disease Neuroimaging Initiative (ADNI). To allow for greater generalisability and validation, Minder and ADNI ADAS-Cog scores were adjusted to reflect the 11-item ADAS-Cog scale. Scores for delayed recall, the maze task, and number cancellation were removed, and the total ADAS-Cog score was recalculated. Table 1 outlines the assessed domains, scoring, and time required for each clinical assessment. Assessments are collected at regular intervals during home visits made by the Minder clinical monitoring team. Figure 1a outlines the timeline of clinical assessment collection in the Minder study.

**Comorbidities.** Information about participants' comorbid medical conditions was extracted from their Electronic Health Records (EHR) and grouped by International Statistical Classification of Diseases and Related Health Problems, 10th revision (ICD-10) chapter. All previous diagnoses recorded in participants' EHRs were included during modelling. Chapters were then converted to 15 binary features indicating presence/absence of each class of disease. Chapters included were: (I) Certain infectious and parasitic diseases, (II) Neoplasms, (III) Diseases of the blood and blood-forming organs and certain disorders involving the immune mechanism, (IV) Endocrine, nutritional and metabolic diseases, (V) Mental and behavioural disorders, (VI) Diseases of the nervous system, (VII) Diseases of the eye and adnexa, (VIII) Diseases of the ear and mastoid process, (IX) Diseases of the circulatory system, (X) Diseases of the respiratory system, (XI) Diseases of the digestive system, (XII) Diseases of the skin and subcutaneous tissue, (XIII) Diseases of the musculoskeletal system and connective tissue, (XIV) Diseases of the genitourinary system, and all other comorbidities.

**Definition of outcome.** Two targets were selected for predictive modelling: MMSE and BADL score at 12 months. First, 12-month rate-of-change values were obtained by subtracting a participant's score at 12 months from their baseline score for both BADL and MMSE. To reduce the impact of outliers on model performance, all 12-month rate-of-change values were winsorised at the 5th and 95th percentiles. Prior to modelling, winsorised rate-of-change values were added to baseline MMSE and BADL scores to produce 12-month scores robust against outliers.

**Data pre-processing.** Because of the COVID-19 pandemic, many home visits for collecting clinical assessments were cancelled or conducted online/on the phone. For this reason, assessment data is missing, only partially complete, or inconsistent across the cohort between 2020 and 2022. This reduced the number of viable clinical assessment trajectories available for this study.

All raw assessment data includes sub-questions and total scores along with assessment-related notes. For each questionnaire, any individual

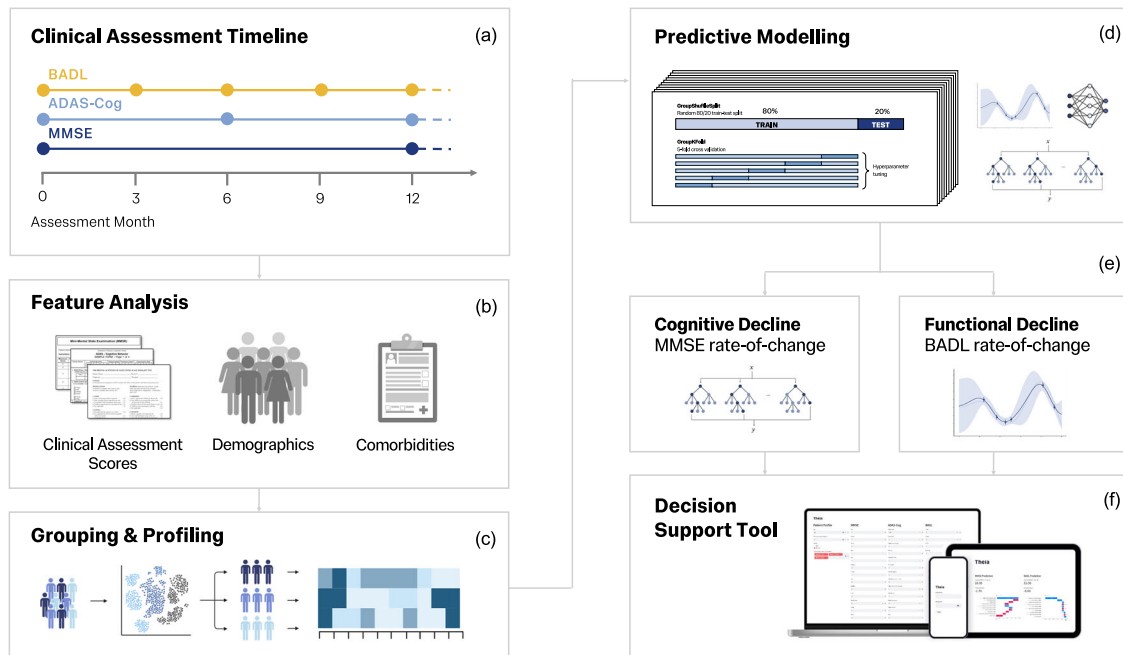

**Fig. 1 | Study design and analysis pipeline.** Clinical assessments are collected at regular intervals throughout the Minder study (**a**), features used for statistical analysis and predictive modelling included clinical assessment scores, participant demographics, and comorbidities (**b**), participants were first grouped based on their relative cognitive and functional decline trajectories and profiled accordingly (**c**), predictive models of cognitive and functional decline were fine-tuned and evaluated using a nested cross-validation approach (**d**), and models were selected and finalised for each outcome measure (**e**). Finally, a decision support tool was designed to deploy both predictive models in clinical settings (**f**). MMSE Mini-mental state exam, ADAS-Cog Alzheimer's Disease Assessment Scale-Cognitive Subscale, BADL Bristol Activities of Daily Living.

assessments with more than 50% missing data and no notes were removed. For any assessments with missing data explained by clinical notes, scores were imputed in accordance with the explanation in the note. Any assessments with notes suggesting data were missing for pathology-independent reasons were removed. Any remaining missing MMSE or ADAS-Cog subcomponents were imputed using maximum likelihood estimation[32]. For BADL, which was collected more frequently than MMSE or ADAS-Cog, any missing values were imputed with a participant's most recent score (collected three months prior). Across assessments, following imputation, all total scores for each questionnaire were re-calculated.

Because scoring across clinical assessments differs, with an increase in ADAS-Cog and BADL and a decrease in MMSE all indicating a decline in performance, all ADAS-Cog and BADL scores were inverted for consistency. Following this inversion, across assessments, lower scores correspond to greater impairment and higher scores to less impairment.

### Statistics and reproducibility
**Unsupervised and group analysis.** Exploratory and statistical analyses were conducted to assess existing relationships in the data and support feature selection and model development. Correlations between baseline assessment scores and 12-month MMSE and BADL rate-of-change were assessed using a two-sided Spearman's rank correlation test[33] with Benjamini-Hochberg False Discovery Rate (FDR) correction for multiple comparisons.

Participants were then divided into 3 groups based on their 12-month MMSE and BADL rate-of-change using K-Means discretisation. Each participant was assigned to the cluster whose centroid (mean rate-of-decline) was nearest, in terms of Euclidean distance, to their rate-of-change, thereby forming "clusters" that reflect the natural groupings in the data[34]. Clusters were labelled based on their average 12-month-MMSE- and BADL-rate-of-change, and participant demographic and clinical profiles were compiled for each cluster. Radar plots were produced to illustrate

differences in average performance across clinical assessment components for each rate-of-change group. To assess the statistical difference between the groups, a Kruskal-Wallis test[35] with Benjamini-Hochberg FDR correction for multiple comparisons was used. Dunn's posthoc pairwise tests for multiple comparisons were then used to identify which specific groups differed significantly from one another. The elbow method was used to identify a suitable range of optimal cluster numbers; from that range, three clusters were selected to balance statistical fit and clinical interpretability.

**Predictive model development.** To predict a participant's MMSE and BADL scores one year in the future using baseline clinical assessment data, comorbidities, age, and sex, several regression models were considered. Performance of each was compared using mean-squared-error (MSE) and mean-absolute-error (MAE) values. The models included $L1$ and $L2$ Regularised Linear Regression, ElasticNet Linear Regression, eXtreme Gradient Boosting Decision Trees (XGBoost)[36], Random Forests[37], Gaussian Process (GP)[38] and a Multilayer Perceptron (MLP).

Regression model performance was evaluated using a nested cross-validation approach (Fig. 1d). The dataset was split into 10 randomised training and testing sets at a ratio of 80:20. For each split, hyperparameters were tuned using a Bayesian or grid search with 5-fold cross-validation. A list of hyperparameter search spaces can be found in the Supplementary table S1. Each model pipeline included a normalisation step followed by feature selection using Recursive Feature Elimination (RFE), with the estimator being set to the model being optimised, and features systematically reduced to half the original feature dimensions. All the categorical features were one-hot-encoded as binary features, and numerical features were standardised using a z-scaler. The model was then re-trained with the best hyperparameters and evaluated on the held-out test set. All data splits were performed using stratification methods to ensure data from the same participant was not split across both sets and that each train-test split preserved the sex distribution across the datasets. The overall model performance for

## Table 2 | Cohort demographics

| | MMSE Model Cohort | BADL Model Cohort | Validation Cohort |
|---|---|---|---|
| 12-month trajectories (n) | 79 | 74 | 741 |
| Participants (n) | 40 | 38 | 379 |
| Female (%) | 21 (52%) | 21 (55%) | 140 (37%) |
| Age at baseline (SD) | 80 (9) | 81 (9) | 75 (7) |
| Ethnicity | | | |
| White | 35 | 33 | 359 |
| Asian | 4 | 4 | 7 |
| Black | 1 | 1 | 13 |
| Household | | | |
| PLWD lives alone | 17 | 21 | N/A |
| PLWD lives with partner | 23 | 17 | N/A |
| Primary diagnosis | | | |
| Alzheimer's Disease | 36 | 35 | 111 |
| Mild Cognitive Impairment | 4 | 3 | 268 |

*MMSE* Mini-Mental State Examination, *BADL* Bristol Activities of Daily Living, *PLWD* Person Living With Dementia.

each regression model was estimated by averaging MSE, MAE, and $R^2$ scores across the 10 splits and calculating 95% Confidence Interval (CI). An ablation study was then performed with BADL scores and comorbidities to assess their necessity as input features. Following model selection and fine-tuning, the best performing model of cognitive decline was re-trained on the entire Minder training dataset and tested on the external ADNI cohort data. Validation performance was evaluated using MSE, MAE, and $R^2$. To visualise feature importance and model decision-making in the best models, Shapley Additive exPlanations (SHAP) values were calculated, and SHAP summary plots were produced.

To address the limitations imposed by our sample size, we applied the approach proposed by Riley et al.[39]. This approach estimates the minimum required sample size for a predictive model with a continuous outcome based on a models' predicted $R^2$, number of parameters, and target mean and standard deviation. The calculations were implemented using the *pmsampsize* package in R. Because our available sample size was fixed, we used this method inversely to determine the maximum number of features that would keep the minimum sample size below our actual sample size. To ensure that our predictions met the criteria outlined in Riley et al.[39], we conducted a larger-scale validation on a dataset obtained from ADNI (n=741) and also investigated how the model training and validation complied with the criteria discussed in their work. Further details on this are included in the Supplementary Results. We have also included a data ablation study to assess the effect of data size on model performance. Results are reported in the Supplementary Results.

**Analysis platform.** All analysis, modelling, and digital tool development were conducted using Python version 3.11.5 and the following associated libraries: Pandas(ver.2.0.3)[40], NumPy(ver.1.24.3)[41], Scikit-learn(ver.1.4.1)[42], and SciPy stats(ver.1.11.3)[43].

### Clinical decision support tool development

A graphical user interface (GUI) was developed using Streamlit(ver.1.32.0)[44], an open-source Python library, which is used to develop computer applications. Streamlit allows for the integration of machine learning algorithms built in Python into web applications. Members of the Minder clinical team were consulted during development of the tool to improve its overall design and usability in clinical settings.

## Results

### Cohort descriptions and demographics

The Minder Health Management Study is a longitudinal follow-up study of PLWD. Data from Minder participants with an AD or MCI diagnosis were used to predict future cognitive and functional ability, as measured by MMSE and BADL scores 12 months in the future. In this study, each 12-month interval of available participant data was considered as an independent trajectory. Participants who were enrolled for more than one year could therefore contribute multiple non-overlapping trajectories, with the first time point in each interval treated as a distinct baseline that included updated assessments and comorbidities. A total of 79 MMSE trajectories from 40 participants (three intervals: $n = 11$, two intervals: $n = 17$, one interval: $n = 12$) were used to model cognitive decline, and 74 BADL trajectories from 38 participants (three intervals: $n = 8$, two intervals: $n = 20$, one interval: $n = 10$) were used to model functional decline. Additionally, an independent dataset comprising 741 12-month trajectories from 379 participants in the Alzheimer's Disease Neuroimaging Initiative (ADNI) study was used for external validation of our model of cognitive decline. Descriptions of the MMSE and BADL model cohorts, as well as the ADNI validation cohort, can be found in Table 2. MMSE and BADL rates of change are reported for their respective cohorts in Table 3 under the "Total" columns and for the ADNI dataset in Table 4. Briefly, the average MMSE rate of change was −1.7 points (SD = 3.0, range −8 to 4) in Minder and −1.3 points (SD = 2.8, range −14 to 7) in ADNI, while the average BADL rate of change was −4.1 points (SD = 5.5, range −17 to 4) in Minder.

### Statistical relationships between model features and MMSE and BADL rate-of-change

Spearman's rank correlation tests revealed a significant correlation between baseline total ADAS-Cog scores and MMSE rate-of-change ($p < 0.05$), but not between baseline total MMSE or BADL scores and MMSE rate-of-change ($p = 0.65$ and $p = 0.18$, respectively). Similarly, both raw baseline ADAS-Cog and MMSE scores were significantly associated with BADL rate-of-change ($p < 0.01$), but baseline BADL score was not, after adjusting for multiple comparisons ($p = 0.07$). Spearman's rank correlation tests were also conducted to assess associations between each clinical assessment subitem, comorbidity class, and demographic feature and MMSE and BADL rates-of-change. All questions significantly correlated with MMSE and BADL rates-of-change are indicated by asterisks in the appropriate radar plots in Fig. 2 (*$p < 0.05$, **$p < 0.01$, ***$p < 0.001$). While features related to wider health status, including demographics and co-morbid health conditions, were considered as model features, we did not observe any significant correlations between these features and MMSE rate-of-change, and found age to be significantly correlated with BADL rate-of-change ($p < .01$).

### Unsupervised and group analysis

Participants were clustered into three groups based on their relative MMSE and BADL 12-month rates-of-change using a K-Means discretisation approach. A complete overview of the demographic profiles of each group, along with their average rate-of-change, can be found in Table 3. In the MMSE model cohort, no baseline assessment scores were significantly different across clusters (MMSE: $p = 0.937$, ADAS-Cog: $p = 0.170$, BADL: $p = 0.307$). In the BADL model cohort, all three baseline assessment scores were significantly different across groups (MMSE: $p < 0.01$, ADAS-Cog: $p < 0.01$, BADL: $p < 0.05$). Significant posthoc pairwise associations between groups for each clinical assessment can be found in Supplementary table S4. Relative performance on each clinical assessment question for each rate-of-change group is visualised in Fig. 2. Asterisks denote which features were significantly correlated with MMSE or BADL rate-of-change and the degree of significance.

## Table 3 | Demographics and baseline clinical assessment scores for each rate-of-change group

| | MMSE rate-of-change | | | |
| --- | --- | --- | --- | --- |
| | Total | Slow | Moderate | Steep |
| n | 79 | 18 | 48 | 13 |
| MMSE rate-of-change (SD) | −1.7 (3.0) | 2.2 (1.3) | −1.8 (1.4) | −6.5 (1.3) |
| Age (SD) | 80 (9) | 80 (9) | 80 (9) | 79 (11) |
| Female (%) | 40 (51%) | 11 (61%) | 24 (50%) | 5 (38%) |
| Diagnosis | | | | |
| AD (%) | 72 (91%) | 15 (83%) | 45 (94%) | 12 (92%) |
| MCI (%) | 7 (9%) | 3 (17%) | 3 (6%) | 1 (8%) |
| Household | | | | |
| PLWD lives alone | 32 (41%) | 8 (44%) | 20 (42%) | 4 (31%) |
| PLWD lives with partner | 47 (59%) | 10 (56%) | 28 (58%) | 9 (69%) |
| Baseline Assessment Scores | | | | |
| MMSE (min-max) | 21.8 (11–30) | 22.3 (13–29) | 21.8 (13–30) | 21.1 (11–30) |
| ADAS-Cog (min-max) | 21.8 (5–56) | 16.8 (8–25) | 22.0 (5–47) | 27.7 (7–56) |
| BADL (min-max) | 12.3 (0–39) | 9.7 (0–26) | 11.8 (0–37) | 17.8 (1–39) |
| | BADL rate-of-change | | | |
| | Total | Slow | Moderate | Steep |
| n | 74 | 34 | 28 | 12 |
| BADL rate-of-change (SD) | 4.1 (5.5) | −0.4 (1.9) | 5.4 (1.5) | 14.2 (2.6) |
| Age (SD) | 81 (9) | 78 (9) | 82 (9) | 86 (6) |
| Female (%) | 39 (53%) | 17 (50%) | 14 (50%) | 8 (67%) |
| Diagnosis | | | | |
| AD (%) | 68 (92%) | 29 (85%) | 28 (100%) | 11 (92%) |
| MCI (%) | 6 (8%) | 5 (15%) | 0 (0%) | 1 (8%) |
| Household | | | | |
| PLWD lives alone | 31 (42%) | 16 (47%) | 10 (36%) | 5 (42%) |
| PLWD lives with partner | 43 (58%) | 18 (53%) | 18 (64%) | 7 (58%) |
| Baseline Assessment Scores | | | | |
| MMSE (min-max) | 21.6 (8–30) | 23.9 (11–30) | 20.7 (13–29) | 17.4 (8–25) |
| ADAS-Cog (min-max) | 22.1 (5–56) | 18.1 (5–56) | 22.9 (13–40) | 31.6 (13–47) |
| BADL (min-max) | 12.6 (0–39) | 9.7 (0–39) | 13.7 (1–35) | 18.7 (1–38) |

%: percent of cluster/cohort.
*MMSE* Mini-Mental State Exam, *BADL* Bristol Activities of Daily Living questionnaire, *ADAS-Cog* 14-item Alzheimer's Disease Assessment Scale-Cognitive Subscale.

## Table 4 | MMSE predictive model and validation datasets

| | Minder | ADNI |
| --- | --- | --- |
| n | 79 | 741 |
| MMSE rate-of-change (SD) | −1.7 (3.0) | −1.3 (2.8) |
| Age (SD) | 80 (9) | 75 (7) |
| Female (%) | 40 (51%) | 251 (34%) |
| Diagnosis | | |
| AD | 72 (91%) | 233 (31%) |
| MCI | 7 (9%) | 508 (69%) |
| Baseline Assessment Scores | | |
| MMSE (min-max) | 21.8 (11–30) | 25.2 (5–30) |
| ADAS-Cog (min-max) | 33.9 (10–76) | 14.0 (1–57) |

*ADNI* Alzheimer's Disease Neuroimaging Initiative, *MMSE* Mini-Mental State Exam, *ADAS-Cog* Alzheimer's Disease Assessment Scale-Cognitive Subscale 11.

provided predictions of future MMSE scores with the lowest average MSE. An ElasticNet Linear Regression model trained on data excluding comorbidities and MMSE scores provided predictions of 12-month BADL rate-of-change with the lowest average MSE. MSE and MAE values for each of the best models are displayed in Table 5 and results from all other models considered can be found in the Supplementary Results. Results from our models trained with reduced feature numbers differed minimally from those presented here and can be found in the Supplementary Results. Additionally, results from a data ablation study can be found in the Supplementary Results.

**External validation.** The model of cognitive decline was further tested on an external validation cohort comprised of participants from the Alzheimer's Disease Neuroimaging Initiative (ADNI) database. While this cohort was overall younger, less female, and less cognitively impaired than the Minder training cohort (Table 4), the model performance was comparable to that of the training set, and MAE was below the standard deviation of each cohort. Results are displayed in Table 5.

### Explanation of model predictions and interpretability
Figure 3 displays SHAP feature importance plots for the best performing models of cognitive (a) and functional (b) decline. For the model that predicts 12-month MMSE score, overall performance on the MMSE and ADAS-Cog at baseline was among the ten most predictive features. Lower baseline scores, indicating greater cognitive impairment, were predictive of lower MMSE scores in 12 months. More specifically, lower scores on ADAS-Cog questions related to ideational praxis, word recall, spoken language, and word recognition, as well as on MMSE questions related to orientation, visuospatial ability, and recall, predicted lower MMSE scores in 12 months. For the model that predicts BADL 12-months in the future, lower impairment on the BADL overall, along with six of its subitems, particularly those related to food and drink preparation, finances, dressing, shopping, and games and hobbies, were all predictive of less overall impairment on the BADL in 12-months. Of the ADAS-Cog questions, word recall and word recognition were most predictive of future ADL ability.

### Demonstrating personalised insights
In Fig. 4, we demonstrate how our models work with unseen data in real-world scenarios. One set of MMSE and BADL trajectories was randomly selected from the study cohort and held out of model training. The selected participant is a male, over age 80 when all baseline assessments were collected. Figure 4 provides individual SHAP waterfall plots for the MMSE and BADL predictions along with actual and predicted MMSE and BADL scores for that participant. In this case, the participant's scores on the MMSE and ADAS-Cog overall, and ideational praxis, spoken language, and word recognition pushed the model's prediction in the negative direction,

### Model performance
L1 and L2 Regularised Linear Regression, ElasticNet Linear Regression, XGBoost[36], Random Forest (RF)[37], Gaussian Processes regression[38] and a Multilayer Perceptron (MLP) were applied to MMSE and BADL trajectory data and performance was compared across models using MSE and MAE. Training and testing splits were stratified to ensure multiple trajectories from the same participant were only included in training or testing sets, but not both. Of the models considered, an ElasticNet linear regression model trained on data including comorbidities and excluding BADL scores

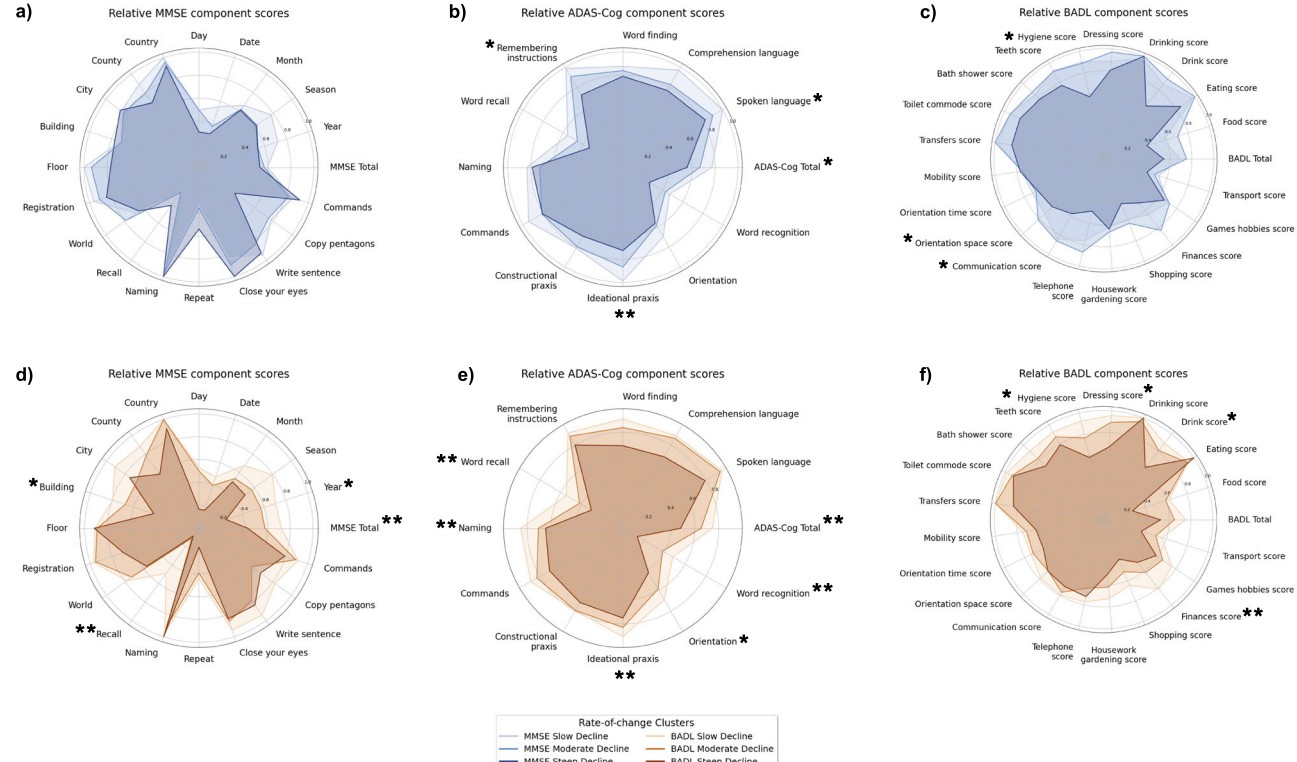

**Fig. 2 | Relative clinical assessment total and component scores by rate-of-change group. a–c** MMSE rate-of-change (n = 79), **d–f** BADL rate-of-change (n = 74). Scores are scaled to a range of 0–1 for the purpose of comparison. Statistical significance, as measured by a two-sided Spearman rank correlation test with Benjamini-Hochberg FDR correction, of the correlation between each question/assessment and the outcome of interest is indicated by asterisks (*p<0.05, **p<0.01). MMSE Mini-Mental State Exam, ADAS-Cog 14-item Alzheimer's Disease Assessment Scale-Cognitive Subscale, BADL Bristol Activities of Daily Living.

### Table 5 | Predictive model performance

| MMSE Model | MSE (95% CI) | MAE (95% CI) | $R^2$ (95% CI) |
|---|---|---|---|
| MMSE + ADAS-Cog | 5.64 (4.40–6.88) | 1.85 (1.65–2.04) | 0.74 (0.67–0.81) |
| MMSE + ADAS-Cog + Comorbidities | **5.60 (4.42–6.78)** | **1.84 (1.64–2.04)** | **0.74 (0.67–0.80)** |
| MMSE + ADAS-Cog + BADL | 7.84 (6.15–9.52) | 2.28 (1.98–2.58) | 0.69 (0.57–0.80) |
| MMSE + ADAS-Cog + BADL + Comorbidities | 7.83 (6.21–9.44) | 2.28 (1.99–2.56) | 0.69 (0.58–0.80) |
| Validation Cohort (ADNI) | 8.01 | 2.19 | 0.69 |
| BADL Model | MSE (95% CI) | MAE (95% CI) | $R^2$ (95% CI) |
| BADL | 25.45 (18.47–32.42) | 3.99 (3.54–4.45) | 0.75 (0.69–0.80) |
| BADL + MMSE | 24.38 (19.48–29.29) | 3.94 (3.55–4.33) | 0.75 (0.71–0.80) |
| BADL + ADAS-Cog | **22.93 (17.87–28.00)** | **3.88 (3.46–4.30)** | **0.77 (0.72–0.82)** |
| BADL + MMSE + ADAS-Cog | 23.96 (18.69–29.23) | 3.93 (3.52–4.35) | 0.76 (0.71–0.81) |
| BADL + MMSE + ADAS-Cog + Comorbidities | 26.64 (20.81–32.47) | 4.11 (3.64–4.58) | 0.74 (0.69–0.78) |

Bolded rows indicate the model with the best performance for each target.

*MMSE* Mini-Mental State Exam, *ADAS-Cog* Alzheimer's Disease Assessment Scale-Cognitive Subscale 11, *BADL* Bristol Activities of Daily Living questionnaire, *MSE* Mean-squared-error, *MAE* Mean-absolute-error.

suggesting that this participant's cognitive function would be lower than the average participant's in 12 months. Only his performance on the Orientation question of the ADAS-Cog shifted the model's prediction toward a slightly better future outcome. In the case of functional decline, our model predicted a higher score than the cohort average for this participant. It is important to note that BADL features remain inverted during modelling, meaning that a lower score is associated with worse performance, but the final target has been reverted to the original scale, so an overall increase in predicted BADL indicates increased impairment. This participant's dressing, finances, hygiene, teeth and food scores on the BADL, along with word

finding and word recognition scores on the ADAS-Cog, all suggested worse ADL outcomes in 12 months. These plots illustrate how each individual prediction is shaped by a patient's specific feature values, rather than being solely influenced by the features with the greatest overall impact across the cohort (as seen in Fig. 3).

### Decision support tool

A decision support tool, Theia, has been developed to deploy both algorithms in clinical settings. Clinicians input MMSE, ADAS-Cog, and BADL scores along with a patient's age, sex, and medical history data. Necessary

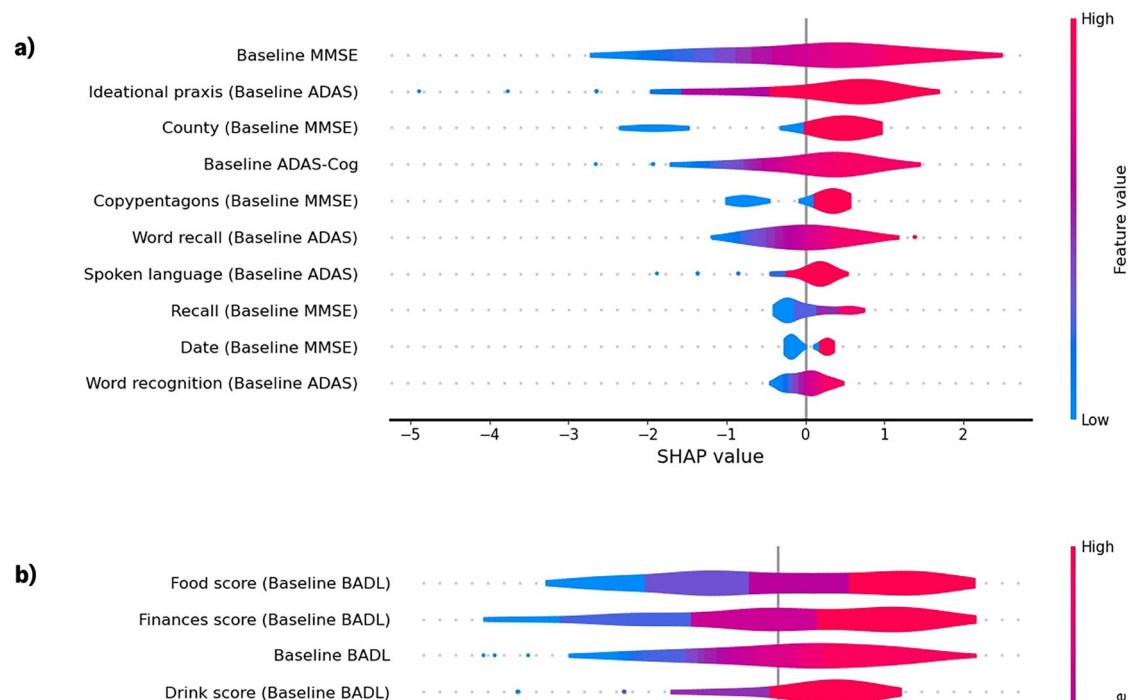

**Fig. 3 | SHapley Additive exPlanations (SHAP) plots for 12-month MMSE and BADL rate-of-change prediction models. a** MMSE predictive model (*n* = 79), **b** BADL predictive model (*n* = 74). SHAP values, shown on the x-axis, indicate how much a given feature influences the predicted rate-of-change value, with positive values driving higher predictions and negative values driving lower predictions. The colour gradient represents the corresponding feature values, demonstrating how specific feature values drive model decision-making. BADL and ADAS-Cog scores are inverted so, across all 3 clinical assessments, a lower score indicates greater impairment.

model features are then input into pre-trained cognitive and functional decline predictive models, and predicted scores are provided. Along with predicted scores, SHAP waterfall plots are created for each model output to describe how input features contributed to the model's prediction. Theia is not available publicly at this time, for the purposes of protecting patient privacy. If you would like to receive access to the tool, please contact the corresponding authors. A schematic of Theia can be found in Fig. 5.

## Discussion

In this study, two predictive models of 12-month decline in dementia were developed using readily available clinical data. One that predicts cognitive decline, as measured by the MMSE, and another that predicts daily functional decline, as measured by the BADL questionnaire, 12 months in the future. The best performing models of future MMSE and BADL scores were both ElasticNet linear regression models which predicted MMSE scores with an MAE of 1.84 (95% CI: 1.64–2.04) and BADL scores with an MAE of 3.88 (95% CI: 3.46–4.30). The two models were designed to require a combination of the same features, allowing them to be used in tandem to support care planning and clinical decision-making.

While both the MMSE and ADAS-Cog are collected for Minder participants, the MMSE was chosen as the outcome measure to predict cognitive decline in this study. ADAS-Cog, the preferred cognitive assessment in many clinical trials, is generally considered to have higher granularity,

reliability, and validity than the MMSE[16]. However, the MMSE remains the most widely used cognitive assessment in clinical settings[16]. Despite considerable debate over the relative benefits and disadvantages of the ADAS-Cog and MMSE in assessing cognitive function and decline in PLWD [16], statistical analysis demonstrated a significant correlation between 12-month-MMSE-rate-of-change and 12-month-ADAS-Cog-rate-of-change, suggesting that the two assessments detect cognitive changes comparably in the Minder cohort.

Groups defined by their relative 12-month MMSE rate-of-change did not differ significantly in any of the three baseline clinical assessments used here, suggesting that total assessment scores alone cannot provide sufficient information about rate of decline in AD/MCI. Further statistical analysis assessing the pairwise association between each questionnaire sub-component score and a participant's 12-month rate-of-change identified three ADAS-Cog and three BADL subquestions as significantly correlated with rate of decline. The number of associations between assessment sub-items and total scores and BADL rate-of-change was even higher. These results prompted the inclusion of assessment subcomponent scores as features in the predictive model developed. Statistical and group-wise analysis provided an important foundation for understanding the data and the broader problem of predicting cognitive and functional decline. However, these approaches are inherently limited to examining associations at the individual feature level and offer only population-level insights.

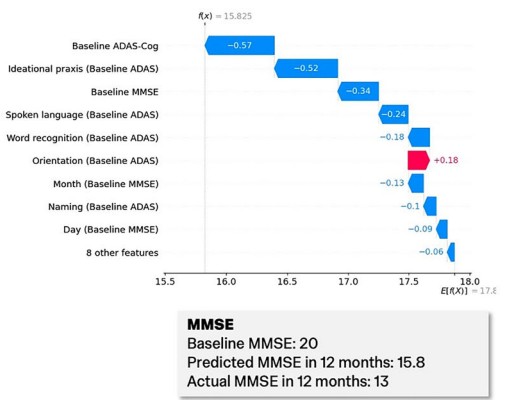

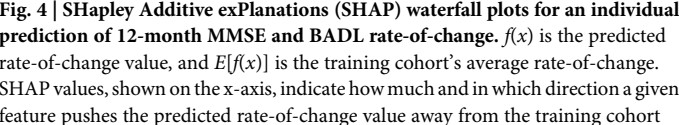

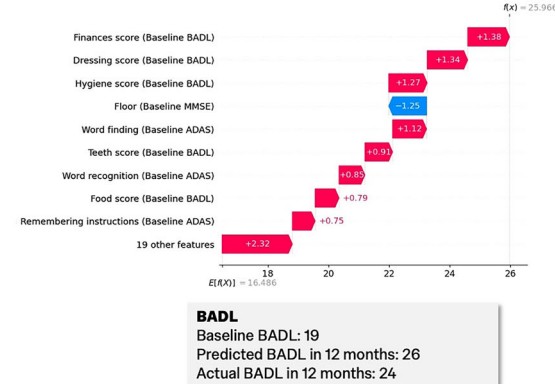

**Fig. 4 | SHapley Additive exPlanations (SHAP) waterfall plots for an individual prediction of 12-month MMSE and BADL rate-of-change.** $f(x)$ is the predicted rate-of-change value, and $E[f(x)]$ is the training cohort's average rate-of-change. SHAP values, shown on the x-axis, indicate how much and in which direction a given feature pushes the predicted rate-of-change value away from the training cohort average, and the waterfall plot illustrates how each feature contributes to the model's overall prediction. Actual and predicted scores are described below each plot. BADL and ADAS-Cog scores were inverted during modelling. The final predicted BADL score has been reverted and thus, a prediction above average indicates greater impairment.

Predictive modelling allowed for exploration of more complex, non-linear interactions between the sub-domains assessed in each questionnaire and for the generation of more precise and personalised individual-level predictions. While statistical analysis describes the trends present across our dataset, predictive modelling captures individual differences in clinical trajectories, thereby offering more personal and clinically relevant insights. Both offer value in the context of dementia care, but the predictive model, in particular, provides a more direct pathway to clinical translation.

Several recent studies have compared the relative sensitivity of clinical assessments, including the MMSE, ADAS-Cog, and BADL, to changes in cognitive, functional, and psychological states along with their diagnostic accuracy[16,27]. Machine learning methods allow us to further investigate the effectiveness of these assessments, not only as diagnostic and monitoring tools but also as predictors of future cognitive and functional states. Baseline total ADAS-Cog score was significantly correlated with MMSE rate-of-change over 12 months, and SHAP analysis revealed that ADAS-Cog and four of its sub-questions were among the 10 most predictive features for the model of MMSE rate-of-change. Specifically, greater impairments in ideational praxis, word recall, spoken language, and word recognition were most predictive of steeper rates of cognitive decline. This is consistent with similar work where overall ADAS-Cog score, along with sub-questions related to word recall, word recognition, and ideational praxis, were found to be highly predictive of cognitive decline as measured by the Clinical Dementia Rating Scale (CDR-SB), a more complex and expertise-dependent cognitive assessment[13]. Given that our model of cognitive decline and the model presented by Devanarayan et al.[13] used different assessments to measure cognitive changes, the similarity in feature importance between the two models is compelling. It further supports the role of the ADAS-Cog, and especially questions related to ideational praxis, word recall, and word recognition as highly predictive of future cognitive function. Further, SHAP analysis confirmed that less impairment on each of these questions is predictive of slower 12-month decline, a finding that cannot be confirmed by the simple feature importance analysis provided in previous studies[13]. Beyond predicting future cognitive assessment scores, there is evidence that the ADAS-Cog is highly a similar, if not better, predictor of conversion from MCI to AD than neuroimaging and fluid biomarkers[45,46]. While many studies have assessed the relative ability of the ADAS-Cog, MMSE, and other cognitive assessments to diagnose and monitor the progression of dementia[16,27], few have assessed their relative predictive power. These findings suggest that, beyond assessing an individual's present cognitive function, the ADAS-Cog may be superior, compared to other cognitive assessments, in predicting the rate of cognitive decline in AD/MCI.

For our model of functional decline, a participant's age, overall baseline BADL score and six specific BADL subitems were among the strongest predictors of future functional ability. In particular, among the top predictors were a participant's ability to prepare food and drink independently and manage their finances. Difficulties with eating and drinking have previously been associated with declines in ADL abilities in PLWD[47,48]. Unlike other ADLs, limitations in these domains can have significant consequences, not only for daily independence, but for the physical health of PLWD. These limitations have been associated with increased hospitalisation rates, which, in turn, are linked to accelerated cognitive and functional decline[47,48]. In addition to ADL, word recall and word recognition, as measured by the ADAS-Cog, were found to be highly predictive of future functional ability. This is supported by Liu-Seifert et al.[23], who found, through statistical analysis, that cognitive assessment scores were highly predictive of functional decline, but ADL scores were not highly predictive of cognitive decline. Liu-Seifert et al.[23] suggest a temporal relationship in which cognitive changes appear before and predict functional changes. Feature importance analysis of both models developed in our study supports Liu-Seifert et al.'s finding. While BADL scores were not necessary for robust prediction of 12-month cognitive decline, ADAS-Cog scores improved BADL model performance and were among the most important predictors of 12-month functional decline.

For both models, ablation studies were performed to evaluate the contributions of specific model inputs on predictive performance. Ablation studies, inspired by a particular technique in neuroscience, involve the removal of a set of related features to evaluate how the model performs in their absence[49]. Here, we assessed the impact of removing BADL and comorbidity features from our models of future cognitive function and different combinations of cognitive and comorbidity features from our models of future functional ability. In the context of clinical prediction, removal of an entire assessment or set of features without a meaningful impact on model performance could improve the generalisability of our models by reducing the time and effort required to make reliable predictions. Some valuable information, however, from particular features within those groups, like a specific ADL or comorbidity, is lost when the entire group of features is removed. Notably, removing BADL-related features improved the performance of the MMSE model, and removing MMSE and comorbidity features improved the performance of the BADL model.

While the association between cognitive and functional decline in PLWD is well documented, there have been mixed findings with respect to the directionality of the relationship. Several studies support a bi-directional

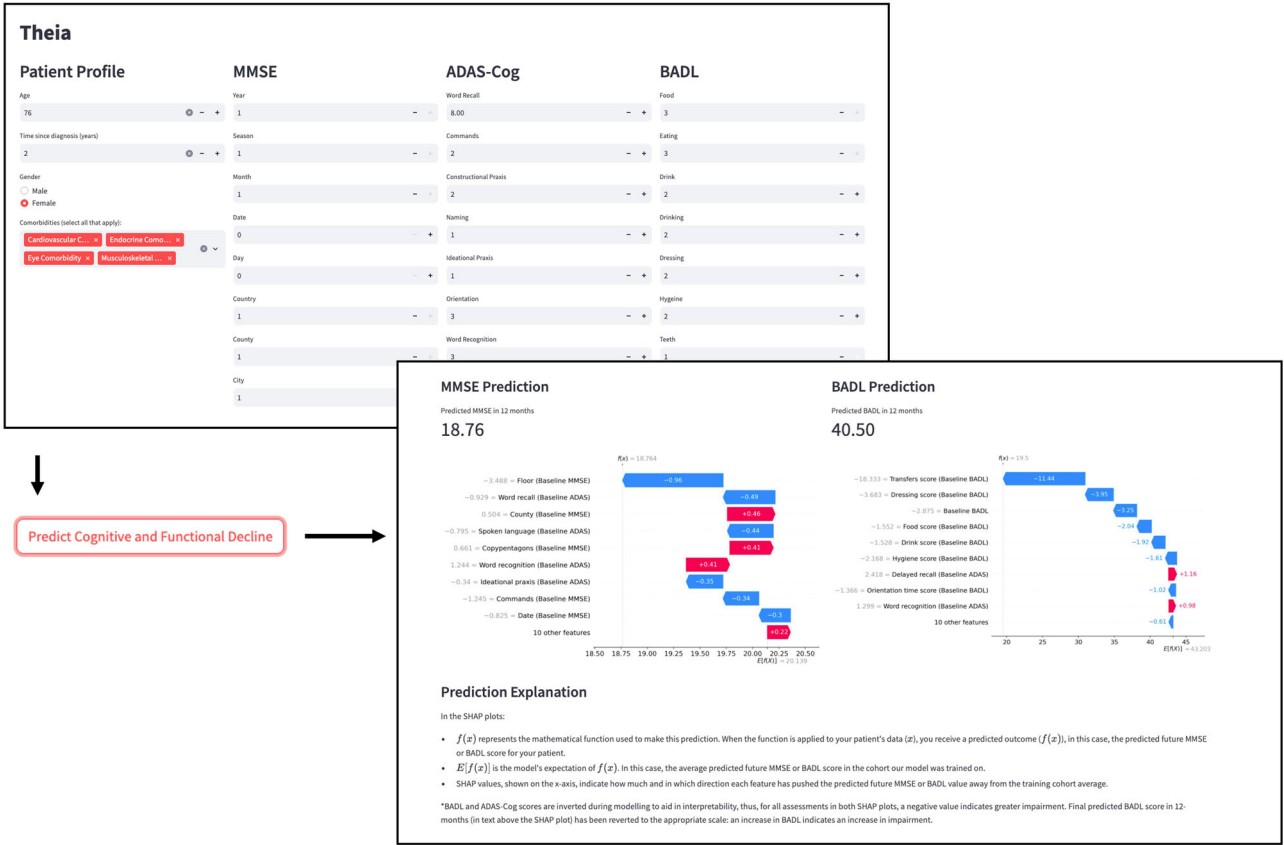

**Fig. 5 | Decision support tool with sample predictions and explanations.** Clinical assessment scores, comorbidity, and demographic data are input by clinicians; necessary model features are extracted and input into the pre-trained predictive models, and a prediction is provided along with SHAP waterfall plots for each prediction to support clinicians in interpreting how predictions were made. The numbers input into the model in this figure are synthetically generated and do not correspond to any particular individual.

association between cognitive and functional decline in PLWD[7,14,50–52], whereas others suggest changes in one domain precede changes in others[23,48,53]. The proposed mechanisms are as follows: (1) consequences of cognitive decline can perpetuate functional decline when, for example, a PLWD forgets to eat or drink or fails to identify early signs of illness, increasing their risk of infections or falls[23,50,53,54], and (2) functional decline often progressively limits a PLWD's ability to engage in physical and social activities, which in turn negatively impacts their cognitive health [47,50]. Our finding that BADL and its sub-components were not required for robust prediction of cognitive decline, but that ADAS-Cog features were necessary for prediction of functional decline, supports the first proposed mechanism for the relationship between these two domains.

According to the 2020 Lancet commission on dementia prevention, intervention, and care, 70–80% of PLWD have at least two additional chronic illnesses, and multimorbidity is associated with a more rapid rate of decline[50,55]. Some studies have found associations between diabetes, cerebrovascular events, cardiovascular comorbidities, cancer and cognitive decline in PLWD[7,51,55], while others suggested sight, oral, and genitourinary comorbidities had the most significant impact on the wellbeing and decline of PLWD[56]. There is clear and substantial evidence for the impact of comorbidities on the health status and wellbeing of PLWD, but there is inconclusive evidence for the role of comorbidities/multimorbidity as predictors of cognitive function, ADLs, and their decline[7,51,55–57]. In our study, no comorbidity features were significantly correlated with rate of cognitive or functional decline, and comorbidities were not among the top predictors of decline during predictive modelling for either model. This finding is supported by a recent study assessing the association between EHR-derived comorbidity data and rate of cognitive decline[19]. Similar to the results of the present study, the authors did not find a strong link between comorbidity

categories and dementia progression[19]. There is compelling evidence for the relationship between multimorbidity and dementia trajectories, though, and this study should not be seen as contradictory to that evidence. In this study, comorbidities were classified based on ICD-10 chapters, which largely separate diseases based on the body system affected. Future work could consider different ways of categorising comorbidity data by, for example, assessing multi-morbidity, severity of diseases, or types of disease. Additionally, because our models are designed to predict cognitive and functional change over a single year, it is possible that the effects of certain comorbidities were not sufficiently observable in that time. As the Minder study progresses and more long-term data are collected, we will assess the role of comorbidities as predictors of longer-term decline. Taken together, these observations could provide valuable insight into mechanistic contributors to cognitive and functional decline in dementia.

Our models were developed with the purpose of facilitating translation into clinical settings and, therefore, rely heavily on clinical assessment data. Given the relative subjectivity of these assessments, though, measurement errors can occur, learning effects are possible, and immediate transient factors like acute illness or normal fluctuations in function during the early stages of dementia might cause variability in the results on some questionnaires[12,58,59]. This has formed the justification for inclusion of more "objective" measures of brain health, including neuroimaging and CSF data in related predictive modelling work[10–15]. In low-resource centres and LMICs, though, where these costly or time-consuming measures of brain health are inaccessible, "pen and paper" neuropsychological assessments are necessary[60,61]. Thus, we sought to provide a proof of concept for the reliability of solely clinical assessment-driven predictive models of decline in dementia, which can be translated to a wide range of clinical settings. While

the MMSE, ADAS-Cog, and BADL questionnaire were used in this study, our aim was less to evaluate these specific instruments and more to demonstrate the feasibility of predicting decline using this type of routinely collected neuropsychological and functional assessment data. Importantly, our models were trained and evaluated on data from individuals with AD or MCI diagnoses and may not generalise to dementia due to other diseases. Dementia can be caused by numerous different diseases, each with a unique physiological and neuropsychological profile[62]. To expand on the work presented here, future studies should assess the predictive ability of clinical assessments in the context of other dementia etiologies.

Although a range of sociodemographic factors were considered during initial predictive modelling, only age and sex were included in the final models. While socioeconomic factors have been shown to influence the rate-of-decline in dementia[63], incorporating features such as employment type, education level, and ethnicity did not lead to a significant improvement in model performance. This may reflect the limited diversity in the Minder study population with respect to these factors, when compared to age and sex. Care should be taken when generalising the findings from this study to people from under-represented sociodemographic groups in our data, as our models have not been directly trained or validated on a sufficiently large amount of data from these populations.

The models developed in this study were designed to predict cognitive and functional change 12 months after baseline assessments are collected, while many other recently published clinical prediction models predict cognitive decline, as measured by several cognitive assessments, 2–4 years in the future[10,12,13]. There is value in predicting outcomes further than 12 months in the future. For example, clinicians may feel less equipped to provide longer-term prognoses and care plans, and may, therefore, be more eager for support in developing them. As participants progress through the Minder study and multi-year data is collected, future work will involve development of models that predict cognitive and functional change further in the future. Nonetheless, cognitive and functional abilities do not change linearly. Understanding how clinical trajectories vary at the monthly and yearly level is valuable as clinicians continue to optimise and personalise care plans in the near-term.

The sample used in this paper is smaller than what is often used to train machine learning models. The Minder study involves close monitoring and involvement by study staff and is, therefore, necessarily smaller than other large-scale longitudinal studies. To account for the small sample, three approaches were taken: dimensionality reduction, nested cross-validation, and external validation. For all models, features were systematically reduced to 50% of the original model dimensions, and model performance was evaluated across ten separate training and testing splits for more robust performance estimation. Additionally, an external validation set was used to test the generalisability of our model of cognitive decline. Our model, trained on the Minder cohort, was applied to unseen and external data from the ADNI dataset, which comprised 741 years of participant trajectories. Performance of our model in the ADNI cohort remained consistent, with a drop in $R^2$ of only 7% from that of the best model tested on Minder data. These findings support the larger generalisability of the models developed in this study. Further external validation of our models that include BADL and comorbidity features could not be completed as no existing longitudinal and publicly available datasets collect this information in addition to MMSE and ADAS-Cog. Our model of cognitive decline improved moderately with the inclusion of comorbidity data, and our model of BADL is an important complement to our model of MMSE.

This paper presents a viable clinical decision support tool to complement and deploy the predictive models developed here. While we have demonstrated proof-of-concept for a set of two predictive models, trained on accessible clinical data, and easily deployed via this tool, substantial future work will be required to assess the utility of the tool in clinical settings. A focus group was consulted during the design of the user interface to aid in its usability and alignment with existing assessment collection systems. Additionally, the inclusion of individualised SHAP plots for each prediction demonstrates the key components driving each predicted outcome, and, along with a detailed description of how to interpret the plots, aids in interpretation by clinicians. Future work will include an assessment of the tool as a decision aid in the clinic.

We have developed a user-friendly decision support tool for use by clinicians that uses real-world clinical assessment data. The tool offers explainable predictive models of 12-month cognitive and functional change in AD/MCI. By integrating it into dementia care planning, this data-driven prognostic support tool can improve the quality of care provided to people living with AD/MCI and offer them a sense of autonomy and control over what is often an unpredictable and challenging future.

## Data availability
Data used in preparation of this article were obtained from the Alzheimer's Disease Neuroimaging Initiative (ADNI) database. As such, the investigators within the ADNI contributed to the design and implementation of ADNI and/or provided data but did not participate in analysis or writing of this report. A complete listing of ADNI investigators can be found here. The Minder data is available from the corresponding author upon reasonable request. Source data for Fig. 2 can be found in Supplementary Data Fig. 2, and source data for Figure 3 can be found in Supplementary Data Fig. 3.

## Code availability
The code for the study and the tool are available at: https://github.com/tmi-lab/theia.

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

## Acknowledgements

This study is funded by the UK Dementia Research Institute (UK DRI) Care Research and Technology Centre funded by the Medical Research Council (MRC), Alzheimer's Research UK, Alzheimer's Society (grant number: UKDRI-7002), and the UKRI Engineering and Physical Sciences Research Council (EPSRC) Resilient Project (grant number: EP/W031892/1). Infrastructure support for this research was provided by the NIHR Imperial Biomedical Research Centre (BRC) and the UKRI Medical Research Council (MRC). Payam Barnaghi is also funded by the Great Ormond Street Hospital and the Royal Academy of Engineering. The funders were not involved in the study design, data collection, data analysis or writing of the manuscript. Data collection and sharing for this project were funded by the Alzheimer's Disease Neuroimaging Initiative (ADNI) (National Institutes of Health Grant U01 AG024904) and DOD ADNI (Department of Defense award number W81XWH-12-2-0012). ADNI is funded by the National Institute on Aging, the National Institute of Biomedical Imaging and Bioengineering, and through generous contributions from the following: AbbVie, Alzheimer's Association; Alzheimer's Drug Discovery Foundation; Araclon Biotech; BioClinica, Inc.; Biogen; Bristol-Myers Squibb Company; CereSpir, Inc.; Cogstate; Eisai Inc.; Elan Pharmaceuticals, Inc.; Eli Lilly and Company; EuroImmun; F. Hoffmann-La Roche Ltd and its affiliated company Genentech, Inc.; Fujirebio; GE Healthcare; IXICO Ltd.; Janssen Alzheimer Immunotherapy Research & Development, LLC.; Johnson & Johnson Pharmaceutical Research & Development LLC.; Lumosity; Lundbeck; Merck & Co., Inc.; Meso Scale Diagnostics, LLC.; NeuroRx Research; Neurotrack Technologies; Novartis Pharmaceuticals Corporation; Pfizer Inc.; Piramal Imaging; Servier; Takeda Pharmaceutical Company; and Transition Therapeutics. The Canadian Institutes of Health Research is providing funds to support ADNI clinical sites in Canada. Private sector contributions are facilitated by the Foundation for the National Institutes of Health (https://www.fnih.org). The grantee organization is the Northern California Institute for Research and Education, and the study is coordinated by the Alzheimer's Therapeutic Research Institute at the University of Southern California. ADNI data are disseminated by the Laboratory for Neuro Imaging at the University of Southern California. We would like to acknowledge support from Surrey and Borders Partnership NHS Foundation Trust and the UK Dementia Research Institute, Care Research and Technology Centre. We are incredibly grateful to the Minder and ADNI study participants for their contributions to this research and the broader research community. **CR&T group**—Acknowledgment list for UK Dementia Research Institute (UK DRI) Care Research & Technology (CR&T) Centre publications using the Minder core data set. The primary contact for this group is: minder-enquiries@imperial.ac.uk *Leadership and Management:* David Sharp (Director), Danielle Wilson (Centre and Research Commercialisation Manager), Sarah Daniels (Hearalth and Social Care Lead), Ramin Nilforooshan (Clinical Lead), David Wingfield (General Practice Lead), Matthew Harrison (Human-Centred Design Lead), Shlomi Haar (Movement Data and Living Lab Lead), Nora Joby (Data Science Lead), Mara Golemme (Scientific Project Manager), Stephanie Lietz (Scientific Project Manager), Margherita Tecilla (Scientific Project Manager). *Behaviour and Cognition Group:* David Sharp (Group Lead), Michael David, Martina Del Giovane, Neil Graham, Magdalena Kolanko, Helen Lai, Lucia M Li, Mark Crook Rumsey, Emma Jane Mallas, Alina-Irina Serban, Eyal Soreq, Abidemi Otaiku, Megan Parkinson, Thomas Parker, Success Fabusoro, Emily Beal, Julian Jeyasingh Jacob, Gaia Frigerio, Anastasia Mirza-Davies, Ethan de Villiers. *Bioelectronic Systems Group:* Timothy Constandinou (Group Lead), Alan Bannon, Danilo Mandic, Ziwei Chen, Charalambos Hadjipanayi, Ghena Hammour, Bryan Hsieh, Amir Nassibi, Adrien Rapeaux, Ian Williams, Maowen Yin, Niro Yogendran. *Robotics and AI Interfaces Group:* Ravi Vaidyanathan (Group Lead), Maria Lima, Ting Su, Melanie Jouaiti, Maitreyee Wairagkar, Carlos Sebastian Castillo, Panipat Wattansiri, Thomas Martineau, Mayue Shi, Tianbo Xu, Alejandro Valdunciel, Reneira Seeamber, Annika Guez, Zehao Liu, Saksham Dhawan, Alina-Irina Serban. *Translational Machine Intelligence Group:* Payam Barnaghi (Group Lead), Nan Fletcher-Lloyd, Samaneh Kouchaki, Alexander Capstick, Chloe Walsh, Louise Rigny, Marirena Bafaloukou, Jin Cui, Yu Chen, Nivedita Bijlani, Iona Biggart, Antigone Fogel, Nathalia Cespedez, Zeinab Ghannam. *Point of Care Diagnostics Group:* Paul Freemont (Group Lead), Michael Crone, Kirsten Jensen, Martin Tran, Thomas Adam, Raphaella Jackson, Alexander Webb, David Wingfield. *Sleep and Circadian Group:* Derk Jan Dijk (Group Lead), Anne Skeldon, Kevin Wells, Ullrich Bartsch, Ciro Della Monica, Kiran KG Ravindran, Victoria L Revell, Hana Hassanin, James Woolley, Iris Wood-Campar, Sarmad Al Gawwam, Aravind Kumar Kamaraj, Marta Pineda Messina. *Brain and Movement Group:* Shlomi Haar (Group Lead), Nathan Steadman, Federico Nardi, Cosima Graef, Alena Kutuzova, Assaf Touboul, Nicolas Calvo Peiro, Jenna Yun, Sean Carr, Uri Rosenblum-Belzer, Emma Burroughs. *Computational Neurology Group:* Gregory Scott (Group Lead), Adela Desowska, Anastasia Gailly de Taurines, Ruxandra Mihai, Nina Moutonnet. *Human Centred Design Group:* Matthew Harrison (Group Lead), Sophie Horrocks, Brian Quan, Victoria Simpson. *Site Investigators and Key Personnel: Surrey and Borders Partnership NHS Foundation Trust:* Ramin Nilforooshan (Chief Investigator), Jessica True (Research and Development Manager), Olga Balazikova (Research and Development Manager), Chloe Walsh (Research Co-ordinator), Nicole Whitethread, Matthew Purnell, Vaiva Zarombaite, Lucy Copps, Olivia Knight, Gaganpreet Bangar, Sumit Dey, Chelsea Mukonda, Jessica Hine, Luke Mallon, Saijal Jhala, Oliver Sargentoni, Amy Alves, Mahan Heydari (Clinical Monitoring Team). *Hammersmith and Fulham Partnership* David Wingfield (Principal Investigator), Monica Morim (Research Nurse/Paramedic), Anesha Patel, Ruby Lyall, Sanara Raza, Success Fabusoro, Gaia Frigerio, Maria Rasulo, Catalina Chavarro Novoa, Martynas Stonkus, Prital Patel, Zara Prem (Clinical Studies), Naomi Hassim, Pippa Kirby (Research Allied Healthcare Professionals), John Patterson(London Borough of Hammersmith and Fulham Support: Assistive Technology), Mike Law (Business Development), Andy Kenny (Social Services). *Minder Digitally Enabled Care for Dementia (MinderCare)* David Sharp (Principal Investigator), James Bird (Co-Investigator), Sarah Pearse (Co-Investigator), Joanna James, Janibo Amade Cassimo, Aglaja Dar, Pandora Wright, Lucia Li, Anastasia Mirza-Davies, Julian Jeyasingh Jacobs, Zinca Zecevic, Sarah Daniels, David Wingfield (Clinical Team), Success Fabusoro, Gaia Frigerio, Maria Rasulo, Catalina Chavarro Novoa (Research Technicians).

## Author contributions

A.F.: Data Collection, Conceptualisation, Methodology, Software, Formal analysis, Investigation, Data Processing, Evaluation, Writing–Original Draft, Review and Editing, Visualisation; C.W.: Data Collection, Conceptualisation, Methodology, Review and Editing; N.F.L.: Conceptualisation, Methodology, Supervision, Review and Editing; CR&T: Data Collection; ADNI: Data Provision; P.M.: Clinical guidance, Supervision, Review and Editing; M.R.: Supervision, Review and Editing; R.N.: Clinical Study Lead, Conceptualisation, Data Collection, Review and Editing, Supervision; P.B.: Conceptualisation, Methodology, Review and Editing, Supervision, Funding Acquisition.

## Competing interests

The authors declare no competing interests.
