## [Transparent Peer Review file · Communications Medicine]

Predicting Rates of Cognitive and Functional Decline in People Living with Alzheimer's Disease and Mild Cognitive Impairment

Corresponding Author: Professor Payam Barnaghi

Version 0:

Reviewer comments:

Reviewer #1

(Remarks to the Author)

I was interested to read the manuscript and appreciated the visualizations throughout the paper. Please see below comments for the authors' consideration:

1. Please clarify how dementia and MCI were diagnosed in the EHR. Were diagnoses made by physicians, neurologists, or based on ICD-10 codes? Additionally, were diagnoses partially based on baseline MMSE or BADL scores? If so, please specify. How many "select cases" were confirmed by MRI or formal cognitive testing?
2. The potential for learning effects with repeated MMSE and BADL assessments could be acknowledged.
3. Participants with AD or MCI and less than 12 months of follow-up were excluded. It would be helpful to describe this group and consider whether their exclusion may bias results, particularly as missing follow-up may reflect disease progression. This would inform to whom the prediction tool is applicable.
4. Consider discuss the generalizability of the tool, particularly given the demographic composition of the Minder cohort, which appears to be predominantly white.
5. Clarify whether the tool is intended for use by clinicians, patients, or both.
6. Were comorbidities identified from the same clinical encounter as the AD/MCI diagnosis, or from any prior EHR data? If a look-back period was used, please specify.
7. " In this study, each year of available participant data was considered. For the model 108 predicting cognitive decline (using MMSE), 79 12-month trajectories from 40 participants were included.." Could authors clarify the rationale for this approach. Would modeling each participant's rate of change per year have been more straightforward? Assuming a constant annual rate of change may be overly simplistic, especially in AD where trajectories may be non-linear.
8. For transparency, consider listing the features used in the model (or at least the types of features considered).
9. I appreciate the authors' effort to justify statistical power. Given the small sample size ($n = 40$) used for the development and validation of a clinical prediction model, would the authors to comment on how their findings align with existing recommendations in the literature regarding minimum sample size requirements for prediction modeling. To my knowledge, current guidance generally suggests a larger sample size for reliable model development and validation. (reference e.g., Riley RD, Snell KIE, Ensor J, et al. Minimum sample size for developing a multivariable prediction model: Part I – Continuous outcomes. *Stat Med.* 2019;38(7):1262–1275. doi:10.1002/sim.7993.)
10. Since the final model was an elastic net linear regression, please consider reporting the estimated beta coefficients.

Reviewer #2

(Remarks to the Author)

This study presents a clinical decision support tool designed to predict rates of cognitive and functional decline in People Living with Dementia (PLWD) at the point of diagnosis. The tool is called Theia, and it is based on two Machine Learning models for dementia trajectory mapping. Their models essentially show that 1) worse cognitive impairment predicts worse cognitive decline and 2) that worse functional impairment (impairment in ADLs and iADLS) predicts worse functional decline in PLWD. The authors indicate that their findings may reshape how the trajectory of decline and care needs in dementia are assessed and planned.

This work is interesting and the use of machine learning models in this context is relatively novel. However, their main finding that worse cognitive and functional impairment predict more rapid cognitive and functional decline is not novel (see citations in first comment below), their findings are based on small (n=40 and N=38) and relatively homogenous samples of individuals of individuals with a clinical diagnosis of MCI or Alzheimer's disease. As such, the results may not generalize to individuals from non-minority groups or individuals that have dementia due to other causes. I have explained this in more detail below.

Broader comments:

1. The finding that worse cognitive and functional impairment predict more rapid cognitive and functional decline is not novel. These findings essentially confirm a large body of existing literature that state that most neurodegenerative conditions progress more rapidly towards the end stages of the diseases. Examples citations that show this include:

- Davis M, O Connell T, Johnson S, Cline S, Merikle E, Martenyi F, Simpson K. Estimating Alzheimer's Disease Progression Rates from Normal Cognition Through Mild Cognitive Impairment and Stages of Dementia. *Curr Alzheimer Res.* 2018;15(8):777-788. doi: 10.2174/1567205015666180119092427. PMID: 29357799; PMCID: PMC6156780.
- Belleville, S., Gauthier, S., Lepage, É., Kergoat, M.-J., & Gilbert, B. (2014). Predicting decline in mild cognitive impairment: A prospective cognitive study. *Neuropsychology*, 28(4), 643–652. <https://doi.org/10.1037/neu0000063>
- Teri L, McCurry SM, Edland SD, Kukull WA, Larson EB. Cognitive decline in Alzheimer's disease: a longitudinal investigation of risk factors for accelerated decline. *J Gerontol A Biol Sci Med Sci.* 1995 Jan;50A(1):M49-55. doi: 10.1093/gerona/50a.1.m49. PMID: 7814789.
- Davis M, O Connell T, Johnson S, Cline S, Merikle E, Martenyi F, Simpson K. Estimating Alzheimer's Disease Progression Rates from Normal Cognition Through Mild Cognitive Impairment and Stages of Dementia. *Curr Alzheimer Res.* 2018;15(8):777-788. doi: 10.2174/1567205015666180119092427. PMID: 29357799; PMCID: PMC6156780.
- Amariglio RE, Grill JD, Rentz DM, Marshall GA, Donohue MC, Liu A, Aisen PS, Sperling RA. Longitudinal Trajectories of the Cognitive Function Index in the A4 Study. *J Prev Alzheimers Dis.* 2024;11(4):838-845. doi: 10.14283/jpad.2024.125. PMID: 39044492; PMCID: PMC11266220.

2. The authors mention that existing models of decline in dementia rely on complex cognitive assessments, neuroimaging, or cerebrospinal fluid data, limiting their scalability in resource-constrained settings, while their models were developed using readily available, clinically validated, and easy-to-administer assessments, enhancing their applicability across clinical settings. However, not using biomarker data (neuroimaging or cerebrospinal fluid) limits the ability of this tool to take the underlying cause of the cognitive and functional decline into account (e.g., alzheimer's, alpha-synucleinopathies, cerebrovascular disease). In addition, it appears that their models were tested on subsets of individuals with MCI (n=4 for the MMSE cohort; n=3 for the BADL cohort) and Alzheimer's disease (n=36 for the MMSE cohort; n=35 for the BADL cohort). Their findings may not generalize to individuals with dementia due to other disease (e.g., alpha-synucleinopathies, cerebrovascular disease) given that each neurodegenerative disease has a different course. The authors' use of the term "People Living with Dementia" to describe this group is somewhat misleading in this context. I suggest "People with a clinical diagnosis of Alzheimer's disease or MCI" as a more accurate term.

3. The authors mention in the abstract that their study is based on a unique longitudinal cohort with multiple repeat assessments. This statement is somewhat misleading given that only 12-month trajectories from 40 participants were included. Specifically, only 11 participants contributed three trajectories, 17 participants contributed two, and 12 contributed one. In addition, 86 participants in their cohort are white and only 13 participants were mixed, Asian, Black, or other. This is a small cohort that may not generalize well to the general population in the UK or the rest of the world. Including more participants including more participants from minority groups and including more timepoints over a longer time span would help the generalizability of the findings.

More specific comments:

- Introduction paragraph 1. The authors claim that a data-driven approach is superior to traditional statistics. However, the rationale for this statement, which is key to this manuscript, remains unclear.
- Introduction line 68 and 69. While the authors cite some manuscripts, these manuscripts do not actually support their claim that the majority of existing models have been developed and evaluated using the same longitudinal Alzheimer's Disease dataset.
- Introduction line 84-88. This is an unnecessarily complex sentence.
- Results line 104. What determined the subset?
- Discussion 353-255. The authors state "While the association between cognitive and functional decline in PLWD is well documented, there have been mixed findings with respect to the directionality of the relationship. Several studies support a bi-directional association between cognitive and functional decline in PLWD, whereas others suggest changes in one domain precede changes in others.

A study that uses a clever longitudinal method to better assess this relationship is:

Zahodne, L. B., Manly, J. J., MacKay-Brandt, A., & Stern, Y. (2013). Cognitive declines precede and predict functional

declines in aging and Alzheimer's disease. PLoS one, 8(9), e73645.

Reviewer #3

(Remarks to the Author)

This is a very well-written and well-constructed manuscript that needs very little editing to be in publishable form. The methods are valid, modern, and thorough. I feel that this paper contributes to the field, and should influence thinking in the field as well. That said, I offer the following minor comments:

General

-I understand the idea behind trying to predict screeners like MMSE/BADL (largely access to data), but it seems like it might be nice to also have a model that accurately predicts cognition past what a screener has to say.

o This is partially addressed in the discussion (3rd paragraph) when they discuss how MMSE and BADL have been shown to accurately represent cognition and ADLs. This should probably be in the Intro section though (or at least mentioned)

-Is there a reason why the Methods section is last instead of before results? It makes for many confusing moments, including when ADAS-Cog is introduced for the first time in the results

-Add to the Discussion a discussion of limitations of this tool, in particular about what happens when the prediction is wrong and that clinical impact. If a clinician wanted to use this tool, what do they need to know about how to describe the predictions to a patient? How "certain" are these predictions?

Specific

-[29-37] how plain should the plain language summary be? This is good for college graduates, but maybe not H.S. graduates. Could simplify more, depending on target audience specified by the journal.

-[69] specify that it's the ADNI dataset, e.g. "...using the same longitudinal Alzheimer's Disease dataset (the Alzheimer's Disease Neuroimaging Initiative)..."

-[68-69] lots of papers cite the ADNI dataset yes, but I think there are also plenty of papers that use a different dataset, so this statement here might not be fully true. And is also not necessary for the point – might be worth removing.

-[105] should it say "every 12 months" here? With methods later in the paper, it was not very clear that there were annual visits here

-[109-111] the word "trajectories" is somewhat confusing here. Are there different study pathways? Using "annual visits" instead might be clearer if that is what this means

-[Table 1] please note any significant differences between groups and associated p-value

-[127-132] This seems to be Methods information rather than Results

-[317-318] cognitive testing for select cases that needed re-evaluation for diagnosis is mentioned. Was this cognitive testing the same as mentioned below in "Clinical Assessments" section? Was it just done by a clinician? Was it a set battery?

-[373-374, 378-379] fairly simple imputation methods were used for some of the missing scores – specifically when the "maximum scores" were used and using the most recent score. Is there a reason why other statistical imputation methods were not used, like multiple imputation or full information maximum likelihood?

o Though I appreciated the steps taken for other missing data that could otherwise be filled in.

Reviewer #4

(Remarks to the Author)

I co-reviewed this manuscript with one of the reviewers who provided the listed reports. This is part of the Communications Medicine initiative to facilitate training in peer review and to provide appropriate recognition for Early Career Researchers who co-review manuscripts.

Reviewer #5

(Remarks to the Author)

The authors present a study utilizing machine learning (ML) models to predict Mini-Mental State Examination (MMSE) and Bristol Activities of Daily Living (BADL) scores after 12 months, leveraging an in-house longitudinal dataset collected from approximately 40 participants over three years. The technical novelty of this work appears to be limited, as it employs an existing ML framework for predictions and explanation. Several major concerns regarding the main claims are outlined below:

Major Concerns:

1. Lack of distinction from prior studies: The authors should cite and clearly differentiate their work from prior studies on prediction models using readily available data sources (e.g., [1-3]), even if the target model outputs differ. This is essential to establish the contribution of the current study.

2. Rationale for target outputs: The rationale behind selecting MMSE and BADL scores as target outputs for supporting care planning remains unclear. Unlike model inputs, model outputs do not necessarily need to be readily collected. Therefore, it is crucial to provide a more detailed explanation for focusing on these scores, rather than other candidates reflecting clinical and neuropathological severities, such as Clinical Dementia Rating (CDR) and biomarkers.

3. Evaluation of the decision support tool: The main claim appears to be the development of a decision support tool, rather than a prediction model, as indicated in the title and throughout the manuscript. However, the tool's usability in clinical practice and its potential to aid dementia care planning have not been evaluated.

4. Limited sample size: The number of participants used in the prediction models is significantly limited. The authors performed a Monte Carlo simulation to justify the sample size by comparing baseline using constant values, but this seems weak. They should consider using other public datasets for external validation or provide a clear explanation of how the dataset and findings in this work are unique and offer new insights in comparison to prior studies, even if the sample size is limited.

Additional Points

5. Analytical details: Several analytical details are unclear to reproduce this work, including: (i) the full list of original feature set before feature selection, (ii) the type and range of hyperparameters for each ML model, (iii) the rationale behind the number of clusters for k-means, and (iv) whether and how the results of unsupervised and group analysis were used in predictive modeling.

6. Statistics and plots: Providing statistics (mean, standard deviation, and range) and plots of changes in MMSE and BADL scores for participants used in predictive modeling would be beneficial.

7. Implication for clinical use in Low- and Middle-Income Countries (LMICs): The authors repeatedly mention LMICs in the manuscript, but the implication of clinical use in these settings is unclear. Specifically, it is essential to explain whether and how the cognitive test scores used as inputs in this tool can be readily available, even in LMICs and low-resource centers.

[1] Yu, Lei, et al. "Predicting age at Alzheimer's dementia onset with the cognitive clock." *Alzheimer's & Dementia* 19.8 (2023): 3555-3562.

[2] Borland, Emma, et al. "Individualized, cross-validated prediction of future dementia using cognitive assessments in people with mild cognitive symptoms." *Alzheimer's & Dementia* 20.12 (2024): 8625-8638.

[3] Adams, Roy, et al. "Clinical factors predicting the rate of cognitive decline in a US memory clinic: An electronic health record study." *Alzheimer's & Dementia: Translational Research & Clinical Interventions* 11.2 (2025): e70070.

Version 1:

Reviewer comments:

Reviewer #1

(Remarks to the Author)

Thank you to the authors for addressing my comments, and in particular for the additional efforts to include ADNI for external validation. I have no further comments. Thank you.

Reviewer #2

(Remarks to the Author)

Reviewer 2

Response to 2.1 and 2.3

While the paragraph is clearly written, the described approach does not constitute a novel contribution. The identification of cognitive subdomains and activities of daily living as predictors of future decline, and the emphasis on subcomponent rather than total scores, are longstanding principles within neuropsychological assessment and research. This reflects the core of what neuropsychology aims to do; link specific cognitive processes to functional outcomes and use those relationships to inform prognosis and care planning. A big limitation I mentioned before is that the models were tested on subsets of individuals with MCI and Alzheimer's disease and findings may not generalize to individuals with dementia due to other disease (e.g., alpha-synucleinopathies, cerebrovascular disease) given that each neurodegenerative disease has a different course. While this is acknowledged in some sections, I still don't think it comes through well that different etiologies are associated with different neuropsychological profiles and that those will predict progression, symptom manifestations, etc. As such, I don't think this manuscript adds much to what is already established by the field of neuropsychology.

Response to 2.2

I appreciate the authors' clarification and efforts to emphasize accessibility and scalability. However, while cost and resource constraints are important considerations, neuroimaging remains essential for characterizing the underlying etiologies of MCI and differentiating between Alzheimer's disease, vascular, and other causes of cognitive decline. Given that the ADNI dataset includes rich imaging data, the authors already have access to modalities that could strengthen their models through etiologic subtyping, even if the predictive focus remains on clinical data.

At present, it is not clear that the proposed tool provides clinically actionable insights beyond what is achievable through existing clinical assessment and neuropsychological approaches. The model's utility would be enhanced if it could help improve/predict diagnostic subtypes or imaging markers (which again is something that the field of neuropsychology does). As it stands, the tool's practical contribution to clinical decision-making remains limited.

Response to 2.4

The authors have now included data from ADNI but they have not described which specific participants from ADNI they have included (e.g., diagnoses, selection process, which ANDI samples) nor have they described patient demographics in the methods.

Response to 2.5. It still remains unclear why the authors believe a data driven approach is superior to traditional statistics

Reviewer #3

(Remarks to the Author)

All of my concerns have been fully addressed.

Reviewer #5

(Remarks to the Author)

Many of my concerns have been addressed in the revised manuscript and the manuscript has improved.

Version 2:

Reviewer comments:

Reviewer #3

(Remarks to the Author)

Reviewer #2 makes points in comments 2.3 and 2.4 that are still not adequately addressed. As stated by Reviewer #2, the authors have access to the ADNI data and while the point of your tool is to provide predictions with clinical data, and you note the strength of ML models in your manuscript and responses, you continue to avoid integration of the neuroimaging data diagnostic features. You cannot have your cake and eat it, too. Training your model with all available data, including the presence of disease etiologies beyond just AD, will strengthen the model when using it solely on clinical data in the future. In other words, include the neuroimaging-based diagnostic etiological data in your training model so that it is more robust to accurate detection when only the clinical testing/self-report data are available. Simply stating that "it was not the focus of the manuscript" is not sufficient, as doing so would make this a unique contribution to the literature that it otherwise is not.

Version 3:

Reviewer comments:

Reviewer #3

(Remarks to the Author)

Thank you for implementing the suggested revision with thoughtful consideration and accurate deployment, and I am glad it was able to strengthen your argument.

I recommend this work for publication.

Dear Editorial team and the reviewers,

We would like to thank the reviewers for their thoughtful and constructive feedback. We have revised the paper in accordance with these comments and provide detailed responses to each comment below.

Original comments have been italicised and shifted to grey; our responses are in black. In the revised manuscript, all changes made are written in blue.

Additionally, we have included a TRIPOD assessment for the AI model.

Thank you again for your time and feedback.

Reviewer #1

I was interested to read the manuscript and appreciated the visualizations throughout the paper. Please see below comments for the authors' consideration:

1.1. Please clarify how dementia and MCI were diagnosed in the EHR. Were diagnoses made by physicians, neurologists, or based on ICD-10 codes? Additionally, were diagnoses partially based on baseline MMSE or BADL scores? If so, please specify. How many "select cases" were confirmed by MRI or formal cognitive testing?

We have revised the manuscript and added a section (S1) to the supplementary information and a brief description in the methods study design & cohort selection section outlining the diagnosis process in community memory clinics. To summarise: diagnoses were made by clinicians in community settings or memory clinics before participants joined the Minder study. While the precise diagnosis process varies slightly between clinicians, the MMSE or MoCA are used to assess cognitive impairment clinically, following the NHS guidelines. It is not standard practice to use the BADL questionnaire during diagnosis and this was included as part of the Minder study protocol. The number of cases where a research diagnosis did not match a clinical diagnosis is now listed in the paper in the methods study design & cohort selection section.

1.2. The potential for learning effects with repeated MMSE and BADL assessments could be acknowledged.

Thank you for this insightful suggestion. This is a common issue in several assessments that are used in clinics. We have revised the discussion section to acknowledge the potential for learning effects in MMSE and BADL assessments.

1.3. Participants with AD or MCI and less than 12 months of follow-up were excluded. It would be helpful to describe this group and consider whether their exclusion may bias

results, particularly as missing follow-up may reflect disease progression. This would inform to whom the prediction tool is applicable.

We have included participants with AD and MCI in the predictions. Those who were excluded did not have a follow up assessment which was the target goal of the predictive assessment model. In other words, we did not have the 12-month MMSE rate of change for some of the participants in the Minder study and those are excluded from the model.

Several reasons could contribute for not having the 12-month follow up assessment; this could be due to participants withdrawing from the research study or not being available to complete the further assessment in the study which included MMSE and BADL. However, we acknowledge that withdrawal or not completing the study assessment within itself could also indicate a subgroup of participants and cause further bias in the study. Our proof-of-concept study focused on assessing the utility of developing a prediction tool with widely available assessment. The reviewer has raised an important point about the application of and potential bias in our models. We have revised the text and added a table (S2) to the supplementary material and a brief description in the methods section that emphasise this.

1.4. Consider discuss the generalizability of the tool, particularly given the demographic composition of the Minder cohort, which appears to be predominantly white.

Thank you for this suggestion. We have taken this into account and have revised the manuscript in two ways. First, we obtained a large validation dataset from the ADNI database and performed an external validation study on it. The results are reported in Table 4 (page 7). We have also reported the demographics of the ADNI participants in Table 1 (page 3). Unfortunately, the ADNI cohort is also predominantly white, which limits our ability to make conclusions regarding our models' performance for ethnically diverse groups. Second, we have revised the discussion section to discuss the barriers to generalisability more explicitly and mentioned requirements to go beyond this proof-of-concept study, i.e. further validation and evaluation in more demographically diverse populations.

1.5. Clarify whether the tool is intended for use by clinicians, patients, or both.

Thank you for raising this. We noticed that this was not explicitly mentioned in the previous version of the manuscript. The tool is designed to be used as decision-support by clinicians in memory clinics or similar settings. The aim is to provide data-driven, personalised, and explainable predictions of cognitive and functional decline, complementing clinicians' insights and intuitions about patient care and support needs during consultations, where baseline assessments are often collected.

We have revised the plain language summary and discussion sections to provide further clarification regarding this.

1.6. Were comorbidities identified from the same clinical encounter as the AD/MCI diagnosis, or from any prior EHR data? If a look-back period was used, please specify.

In the comorbidities section of the study methods, we describe that comorbidities were identified from prior EHR data. We have revised the text to clarify that there was no look-back period or cut-off. All reported comorbidities in participants' EHRs are included.

1.7. " In this study, each year of available participant data was considered. For the model predicting cognitive decline (using MMSE), 79 12-month trajectories from 40 participants were included.." Could authors clarify the rationale for this approach. Would modeling each participant's rate of change per year have been more straightforward? Assuming a constant annual rate of change may be overly simplistic, especially in AD where trajectories may be non-linear.

Thank you for raising this. MMSE data is collected once a year in the Minder study. Our aim has been to provide a prediction of the annual rate of change and update it each time a new baseline is collected. The reviewer's observation is completely right that, within a year, cognitive and functional changes will not be linear; the aim here has been to assist clinicians in anticipating the trend of decline and provide information for care planning.

Minder study collects further in-home activity, sleep and physiological measurement data that we have used in other studies to assess higher granular variations in health and activities of people living with dementia in the Minder study, including:

<https://doi.org/10.1016/j.eclinm.2024.103032>; <https://doi.org/10.1016/j.artmed.2024.102821>; <https://doi.org/10.1038/s41746-023-00995-5>.

To clarify this, we have revised the text and included further discussion on the nature of symptom progression and the timelines in which we measure them.

1.8. For transparency, consider listing the features used in the model (or at least the types of features considered).

We have added three tables (S4, S5 and S6) to the supplementary materials, which list all features considered by each model as well as the features selected by recursive feature elimination (RFE) as part of the pipeline.

1.9. I appreciate the authors' effort to justify statistical power. Given the small sample size (n = 40) used for the development and validation of a clinical prediction model, would the authors to comment on how their findings align with existing recommendations in the literature regarding minimum sample size requirements for prediction modeling. To my knowledge, current guidance generally suggests a larger sample size for reliable model development and validation. (reference e.g., Riley RD, Snell KIE, Ensor J, et al. Minimum sample size for developing a multivariable prediction model: Part I – Continuous outcomes. Stat Med. 2019;38(7):1262–1275. doi:10.1002/sim.7993.).

We acknowledge that the sample size to train the main model is relatively small. To avoid overfitting, we applied stratified and nested cross-validation and reported the confidence intervals. We also conducted a data ablation study to demonstrate the impact of varying data proportions on model training. This study revealed convergence with the reported performance measures in confidence intervals.

We also followed the reviewer's recommendation based on Riley RD, Snell KIE, Ensor J, et al. Minimum sample size for developing a multivariable prediction model: Part I – Continuous outcomes. *Stat Med.* 2019;38(7):1262–1275. doi:10.1002/sim.7993. In a new experiment, we ran our model with the top 13/14 features and reported the results in Supplementary Tables S12 and S13. These results comply with the criteria 1 and 2 stated in Riley RD, *et al.*

To validate the model for criteria 3 and 4 as stated in Riley RD, et al., and to assess its generalisation, we obtained a validation dataset from ADNI (n=741). The evaluation results for this are presented in Table 4. However, since there are no other studies or data available with repeat BADL assessments, we are unable to validate and test the BADL model in a similar manner. We have also revised the discussion section, highlighting the need for further studies and evaluation to assess the generalisability of this proof-of-concept work.

1.10. Since the final model was an elastic net linear regression, please consider reporting the estimated beta coefficients.

We have revised the text. This information can now be found in supplementary tables S4-S6. We have shown all the features in these tables and included the beta coefficients for the features that were included in the predictive models.

Reviewer #2

This study presents a clinical decision support tool designed to predict rates of cognitive and functional decline in People Living with Dementia (PLWD) at the point of diagnosis. The tool is called Theia, and it is based on two Machine Learning models for dementia trajectory mapping. Their models essentially show that 1) worse cognitive impairment predicts worse cognitive decline and 2) that worse functional impairment (impairment in ADLs and iADLs) predicts worse functional decline in PLWD. The authors indicate that their findings may reshape how the trajectory of decline and care needs in dementia are assessed and planned. This work is interesting and the use of machine learning models in this context is relatively novel. However, their main finding that worse cognitive and functional impairment predict more rapid cognitive and functional decline is not novel (see citations in first comment below), their findings are based on small (n=40 and N=38) and relatively homogenous samples of individuals of individuals with a clinical diagnosis of MCI or Alzheimer's disease. As such, the results may not generalize to individuals from non-minority groups or individuals that have dementia due to other causes. I have explained this in more detail

below.

Broader comments:

2.1. The finding that worse cognitive and functional impairment predict more rapid cognitive and functional decline is not novel. These findings essentially confirm a large body of existing literature that state that most neurodegenerative conditions progress more rapidly towards the end stages of the diseases. Examples citations that show this include:

- *Davis M, O Connell T, Johnson S, Cline S, Merikle E, Martenyi F, Simpson K. Estimating Alzheimer's Disease Progression Rates from Normal Cognition Through Mild Cognitive Impairment and Stages of Dementia. Curr Alzheimer Res. 2018;15(8):777-788. doi: 10.2174/1567205015666180119092427. PMID: 29357799; PMCID: PMC6156780.*
- *Belleville, S., Gauthier, S., Lepage, É., Kergoat, M.-J., & Gilbert, B. (2014). Predicting decline in mild cognitive impairment: A prospective cognitive study. Neuropsychology, 28(4), 643–652. <https://doi.org/10.1037/neu0000063>*
- *Teri L, McCurry SM, Edland SD, Kukull WA, Larson EB. Cognitive decline in Alzheimer's disease: a longitudinal investigation of risk factors for accelerated decline. J Gerontol A Biol Sci Med Sci. 1995 Jan;50A(1):M49-55. doi: 10.1093/gerona/50a.1.m49. PMID: 7814789.*
- *Amariglio RE, Grill JD, Rentz DM, Marshall GA, Donohue MC, Liu A, Aisen PS, Sperling RA. Longitudinal Trajectories of the Cognitive Function Index in the A4 Study. J Prev Alzheimers Dis. 2024;11(4):838-845. doi: 10.14283/jpad.2024.125. PMID: 39044492; PMCID: PMC11266220.*

We appreciate the reviewer's thoughtful suggestions on other recent papers similar to our work. It is true that our feature importance analysis found total MMSE and ADAS-Cog to be an important feature in prediction of cognitive decline and total BADL to be highly predictive of functional decline. However, in our study we aim to create personalised and explainable models that explain heterogeneity in these predictions. These models highlight the key features that influence the decline predictions beyond the baseline scores.

In our study, we assess the role of specific sub-domains of cognition and activities of daily living as predictors of future decline and identify areas of focus for clinicians as they collect clinical assessments and prepare care plans for their patients. The results of unsupervised and statistical analysis underscore this. Participants' baseline MMSE scores were not statistically different across rate of change groups, and scores on specific subcomponents of the included assessments predicted decline more strongly than certain total scores. Due to the latter, we present models with explainable features to create more precise and personalised predictions at the individual level.

2.2. The authors mention that existing models of decline in dementia rely on complex cognitive assessments, neuroimaging, or cerebrospinal fluid data, limiting their scalability in resource-constrained settings, while their models were developed using readily available, clinically validated, and easy-to-administer assessments, enhancing their applicability across clinical settings. However, not using biomarker data (neuroimaging or cerebrospinal fluid)

limits the ability of this tool to take the underlying cause of the cognitive and functional decline into account (e.g., Alzheimer's, alpha-synucleinopathies, cerebrovascular disease).

We appreciate the reviewer raising this, as this highlights an important feature of this work in relation to the published studies. We agree that the gold standard for diagnosis and monitoring of dementia is currently imaging and fluid biomarkers. As these technologies progress, better and more scalable techniques will become available. However, these technologies are not widely available in all settings, and they are costly and resource intensive. In this work, our aim has been two-fold: first, to develop a model that can be used by clinicians at the time of consultation to predict patient care needs, and second, to address the inaccessibility of the advanced tools and biomarkers and develop a low-cost and scalable tool that can be used in low-resource settings to complement clinical decision-making.

We have revised the discussion section, highlighted the reviewer's suggestions, and provided further clarification on the utility and limitations of our work.

2.3 In addition, it appears that their models were tested on subsets of individuals with MCI (n=4 for the MMSE cohort; n=3 for the BADL cohort) and Alzheimer's disease (n=36 for the MMSE cohort; n=35 for the BADL cohort). Their findings may not generalize to individuals with dementia due to other disease (e.g., alpha-synucleinopathies, cerebrovascular disease) given that each neurodegenerative disease has a different course. The authors' use of the term "People Living with Dementia" to describe this group is somewhat misleading in this context. I suggest "People with a clinical diagnosis of Alzheimer's disease or MCI" as a more accurate term.

This is an important point; we thank the reviewer for raising it. We have adjusted the wording, where appropriate, throughout the manuscript to reflect the subsets of dementia included in this study. Where we discuss broader population-level data and related work, we have kept the "PLWD" and "dementia" language constant, but wherever we discuss conclusions to be made from our work, wording has been adjusted appropriately.

We have also included a new section in the revised manuscript with an evaluation on a larger cohort from ADNI data. For this validation, we have also reported the diagnosis and the demographics of the cohort.

2.4. The authors mention in the abstract that their study is based on a unique longitudinal cohort with multiple repeat assessments. This statement is somewhat misleading given that only 12-month trajectories from 40 participants were included. Specifically, only 11 participants contributed three trajectories, 17 participants contributed two, and 12 contributed one. In addition, 86 participants in their cohort are white and only 13 participants were mixed, Asian, Black, or other. This is a small cohort that may not generalize well to the general population in the UK or the rest of the world. Including more participants including more participants from minority groups and including more timepoints over a longer time span would help the generalizability of the findings.

Thank you for highlighting this. We have now also included a larger dataset from ADNI (n=741) to validate the model. We have also revised the abstract and discussion section to reflect the demographic and sample limitations of this proof-of-concept study. Please also refer to our response to reviewer 1's comment (1.4).

More specific comments:

2.5 Introduction paragraph 1. The authors claim that a data-driven approach is superior to traditional statistics. However, the rationale for this statement, which is key to this manuscript, remains unclear.

Thank you for highlighting this. After a careful review, a more precise word would be "heuristics", and not statistics (as we also use statistics and statistical learning in this study). We have revised the text to correctly reflect the intention of the tool. Thank you again for noticing this.

2.6 Introduction line 68 and 69. While the authors cite some manuscripts, these manuscripts do not actually support their claim that the majority of existing models have been developed and evaluated using the same longitudinal Alzheimer's Disease dataset.

We have chosen to remove this sentence as this reviewer and reviewer #3 highlight that it is not necessary or very precise.

2.7 Introduction line 84-88. This is an unnecessarily complex sentence.

This sentence has now been segmented and rewritten to improve clarity. Thank you for pointing this out.

2.8 Results line 104. What determined the subset?

The Minder study includes participants with any dementia diagnosis, TBI, stroke, or frailty. For our study, we only included (a "subset" of) Minder participants with an AD or MCI diagnosis. We have revised the text to clarify this.

2.9 Discussion 353-255. The authors state "While the association between cognitive and functional decline in PLWD is well documented, there have been mixed findings with respect to the directionality of the relationship. Several studies support a bi-directional association between cognitive and functional decline in PLWD, whereas others suggest changes in one domain precede changes in others.

A study that uses a clever longitudinal method to better assess this relationship is: Zahodne, L. B., Manly, J. J., MacKay-Brandt, A., & Stern, Y. (2013). Cognitive declines precede and predict functional declines in aging and Alzheimer's disease. PloS one, 8(9), e73645.

This is a very well-aligned study with our research findings. Thank you for sharing it with us. We have cited and briefly mentioned this study in the discussion section.

Reviewer #3

This is a very well-written and well-constructed manuscript that needs very little editing to be in publishable form. The methods are valid, modern, and thorough. I feel that this paper contributes to the field, and should influence thinking in the field as well. That said, I offer the following minor comments:

General

3.1 I understand the idea behind trying to predict screeners like MMSE/BADL (largely access to data), but it seems like it might be nice to also have a model that accurately predicts cognition past what a screener has to say.

- This is partially addressed in the discussion (3rd paragraph) when they discuss how MMSE and BADL have been shown to accurately represent cognition and ADLs. This should probably be in the Intro section though (or at least mentioned).*

Thank you for this suggestion. We have now revised the introduction section to reflect this.

1.3 Is there a reason why the Methods section is last instead of before results? It makes for many confusing moments, including when ADAS-Cog is introduced for the first time in the results

Nature Communications Medicine requires that manuscripts be submitted in this order. We follow the nature guidelines in preparing the manuscript.

2.3 Add to the Discussion a discussion of limitations of this tool, in particular about what happens when the prediction is wrong and that clinical impact. If a clinician wanted to use this tool, what do they need to know about how to describe the predictions to a patient? How “certain” are these predictions?

We have revised the introduction and discussion section, clarified the limitations of the tool and its impact, considering that it has not been currently integrated into clinical practice. This is a proof-of-concept study, and our tool will certainly require further validation and regulatory approval before being suitable for use in real-world clinical decision-making.

Specific 3.4[29-37] how plain should the plain language summary be? This is good for college graduates, but maybe not H.S. graduates. Could simplify more, depending on target audience specified by the journal.

We have revised the plain language summary, following the nature guidelines. We have avoided any technical jargon and specialised language in the descriptions.

3.5[69] specify that it's the ADNI dataset, e.g. "...using the same longitudinal Alzheimer's Disease dataset (the Alzheimer's Disease Neuroimaging Initiative)..."

Please see the response to 3.6 below.

3.6[68-69] lots of papers cite the ADNI dataset yes, but I think there are also plenty of papers that use a different dataset, so this statement here might not be fully true. And is also not necessary for the point – might be worth removing.

Thank you for noticing this. We have decided to remove this sentence.

3.7[105] should it say "every 12 months" here? With methods later in the paper, it was not very clear that there were annual visits here

We have updated this sentence to improve clarity.

3.8[109-111] the word "trajectories" is somewhat confusing here. Are there different study pathways? Using "annual visits" instead might be clearer if that is what this means

We have added clarification in an earlier sentence for this, at the very beginning of the Results section. A trajectory refers to a period of 12 months for which we have all baseline assessment data and an MMSE or BADL score at 12 months after baseline.

3.9[Table 1] please note any significant differences between groups and associated p-value

Thank you for this suggestion. We have revised the text. This has been added to the results section. We have also included a new table in the supplementary material (Table S2) to report p-values for statistically significant pairwise associations between the groups.

3.10[127-132] This seems to be Methods information rather than Results.

We have revised the text and moved this information from the Results section to Methods.

3.11[317-318] cognitive testing for select cases that needed re-evaluation for diagnosis is mentioned. Was this cognitive testing the same as mentioned below in "Clinical Assessments" section? Was it just done by a clinician? Was it a set battery?

Thank you for highlighting this. We noticed that the initial descriptions did not provide sufficient clarity on this point. We have revised the text and provided further elaboration. The revised text is in the methods section and also supplementary section S1.

3.12[373-374, 378-379] fairly simple imputation methods were used for some of the missing scores – specifically when the "maximum scores" were used and using the most recent score.

Is there a reason why other statistical imputation methods were not used, like multiple imputation or full information maximum likelihood?

- *Though I appreciated the steps taken for other missing data that could otherwise be filled in.*

While this imputation technique was selected based on discussions with the Minder clinical team, we agree that it is fairly simple. We have followed the reviewer's suggestion and performed a maximum likelihood imputation. This did not change the predictive model performance. We would also like to clarify that only one sample in our training set was imputed. We have revised the text and reported the results with maximum likelihood imputation in the main text.

Reviewer #5

The authors present a study utilizing machine learning (ML) models to predict Mini-Mental State Examination (MMSE) and Bristol Activities of Daily Living (BADL) scores after 12 months, leveraging an in-house longitudinal dataset collected from approximately 40 participants over three years. The technical novelty of this work appears to be limited, as it employs an existing ML framework for predictions and explanation. Several major concerns regarding the main claims are outlined below:

Major Concerns:

5.1. Lack of distinction from prior studies: The authors should cite and clearly differentiate their work from prior studies on prediction models using readily available data sources (e.g., [1-3]), even if the target model outputs differ. This is essential to establish the contribution of the current study.

*[1] Yu, Lei, et al. "Predicting age at Alzheimer's dementia onset with the cognitive clock." *Alzheimer's & Dementia* 19.8 (2023): 3555-3562.*

*[2] Borland, Emma, et al. "Individualized, cross-validated prediction of future dementia using cognitive assessments in people with mild cognitive symptoms." *Alzheimer's & Dementia* 20.12 (2024): 8625-8638.*

*[3] Adams, Roy, et al. "Clinical factors predicting the rate of cognitive decline in a US memory clinic: An electronic health record study." *Alzheimer's & Dementia: Translational Research & Clinical Interventions* 11.2 (2025): e70070.*

Thank you for suggesting these additional relevant studies. We have revised the introduction and discussion sections to describe the findings of these studies and cited them in the text.

5.2. Rationale for target outputs: The rationale behind selecting MMSE and BADL scores as target outputs for supporting care planning remains unclear. Unlike model inputs, model outputs do not necessarily need to be readily collected. Therefore, it is crucial to provide a more detailed explanation for focusing on these scores, rather than other candidates reflecting clinical and neuropathological severities, such as Clinical Dementia Rating (CDR) and biomarkers.

MMSE is a measure that is often collected in memory clinics in the NHS settings. We rely on the data available via the Minder study for training and testing the models and ADNI data (MMSE) for external validation. We acknowledge that other measures and scales are also available. In this work, our aim has been to demonstrate the utility of using such scales to assist clinical decision-making and care provisioning. The approach and methods proposed in this work can potentially be applied to other scales where data could be available. We have revised the discussion section to briefly highlight this.

5.3. Evaluation of the decision support tool: The main claim appears to be the development of a decision support tool, rather than a prediction model, as indicated in the title and throughout the manuscript. However, the tool's usability in clinical practice and its potential to aid dementia care planning have not been evaluated.

Thank you for highlighting this. We have revised the title to reflect this. The aim of the study has been to develop predictive models and then integrate them into a proof-of-concept tool that can potentially be used in clinical settings. We have also revised the discussion section to reflect the limitations, focus, and utility of this study and the models/tool.

5.4. Limited sample size: The number of participants used in the prediction models is significantly limited. The authors performed a Monte Carlo simulation to justify the sample size by comparing baseline using constant values, but this seems weak. They should consider using other public datasets for external validation or provide a clear explanation of how the dataset and findings in this work are unique and offer new insights in comparison to prior studies, even if the sample size is limited.

We have revised the text and changed the Monte Carlo simulation to a data ablation study showing the changes in models' performance and confidence intervals while using different proportions of the data. The results are included in the supplementary material section S7. We have also included a new validation with a larger dataset from the ADNI database (n=741). We have revised the text and also included the results of this external validation in Table 4 in the results section.

We have also provided a detailed description in relation to this and in response to reviewer 1's comment (1.9).

Additional Points

5.5. Analytical details: Several analytical details are unclear to reproduce this work, including:

- i. the full list of original feature set before feature selection*

We have added three tables to the supplementary information (S4-S6), two that list the original feature set for the MMSE models (with and without comorbidities), and one for the BADL model. In each, the selected features are bolded, and their respective beta coefficients for the ElasticNet model are provided.

ii. *the type and range of hyperparameters for each ML model*

The type and range of hyperparameters for each model are now listed in the supplementary section (section S4).

iii. *The rationale behind the number of clusters for k-means*

We used the elbow method to identify a range of optimal cluster numbers and selected 3 clusters based on clinical utility and interpretability. We have revised the text in the method section to clarify this.

iv. *Whether and how the results of unsupervised and group analysis were used in predictive modeling.*

We have revised the discussion section to clarify this. We have also updated the results and methods section in relation to this. Overall, as we demonstrated in our clustering results, the profiles of the clusters, while providing information at the group level, contain heterogeneity at the individual level. This has been the motivation for developing further predictive models that can also provide explanations at the individual level. We have also provided more details regarding this in response to Reviewer #2's comment 2.1 (second paragraph)

5.6. Statistics and plots: Providing statistics (mean, standard deviation, and range) and plots of changes in MMSE and BADL scores for participants used in predictive modeling would be beneficial.

While mean \pm SD for changes in MMSE and BADL scores in each cohort are provided in Table 2, we have added a brief description directly referencing the table and added the mean, SD and range to the main text for each cohort's rate of change. We have also changed the title of the row in Table 2 from "Slope" to 'MMSE rate of change'/'BADL rate of change' for clarity and placed SDs in parentheses following means instead of \pm .

5.7. Implication for clinical use in Low- and Middle-Income Countries (LMICs): The authors repeatedly mention LMICs in the manuscript, but the implication of clinical use in these settings is unclear. Specifically, it is essential to explain whether and how the cognitive test scores used as inputs in this tool can be readily available, even in LMICs and low-resource centers.

Thank you for highlighting this. We acknowledge that this is an important feature that requires further clarification. While our models have not been directly tested in LMIC settings, we wanted to highlight the potential scalability and lower cost of the assessments required in this work. Compared with more expensive and resource-intensive techniques such as PET, MRI or fluid biomarkers, these baseline methods can be administered with trained staff in community settings. While these assessments are not an alternative for gold standard imaging and fluid biomarkers for diagnosis and progression monitoring, they can provide

clinically applicable insights that can be obtained more widely and at a lower cost for clinical consultations and care planning. We have revised the discussion section to reflect the utility and limitations of the study.

Dear Editorial team and reviewers,

Thank you for taking the time to review the revised version of this manuscript. We have provided a second revised version of the manuscript to address the remaining comments from Reviewer #2.

As with our previous revisions, Original comments have been italicised and shifted to grey; our responses are in black. In the revised manuscript, all changes made are written in blue.

2.1 While the paragraph is clearly written, the described approach does not constitute a novel contribution. The identification of cognitive subdomains and activities of daily living as predictors of future decline, and the emphasis on subcomponent rather than total scores, are longstanding principles within neuropsychological assessment and research. This reflects the core of what neuropsychology aims to do; link specific cognitive processes to functional outcomes and use those relationships to inform prognosis and care planning.

Thank you for highlighting this. We agree that the conceptual link between cognitive subdomains and functional outcomes is a cornerstone of neuropsychology. While the principles underlying neuropsychological assessment (linking cognitive subdomains and functional abilities) are well established, our contribution is not in reiterating these principles but in developing a predictive modelling framework that can scale to real-world clinical decision-support. The goal is to integrate instruments that are already central to dementia care into a data-driven framework, rather than introducing new or less commonly used tests. The MMSE, ADAS-Cog, and BADL are primarily used for monitoring progression, not prognosis: clinicians are advised to make decisions based on heuristics and experience, which, as discussed in the paper, fail to capture individual variability. In contrast, machine learning allows for consideration of complex interactions among assessment components, combinations that number, at minimum, in the order of 10^{14} possible feature configurations (shown below), far beyond what can be evaluated by human reasoning alone. This approach transforms widely used clinical tools into scalable individualised prognostic aids.

MMSE Model Feature Combinations

MMSE	$10 \times 5 \times 3^4 \times 2 \times 1 = 8,100$	}	1.215×10^{14}	}	1.991×10^{18}	}	6.941×10^{27}
ADAS-Cog	$12 \times 10 \times 8 \times 5^8 = 375,000,000$						
Sex	$= 2$						
Age	$= 20$						
Comorbidities	$2^{14} = 16,384$						
BADL	$3^{20} = 3,486,784,401$						

BADL Model Feature Combinations

BADL	$3^{20} = 3,486,784,401$	}	5.230×10^{19}	}	4.236×10^{23}	}	6.941×10^{27}
ADAS-Cog	$12 \times 10 \times 8 \times 5^8 = 375,000,000$						
Sex	$= 2$						
Age	$= 20$						
MMSE	$10 \times 5 \times 3^4 \times 2 \times 1 = 8,100$						
Comorbidities	$2^{14} = 16,384$						

2.2 A big limitation I mentioned before is that the models were tested on subsets of individuals with MCI and Alzheimer's disease and findings may not generalize to individuals with dementia due to other disease (e.g., alpha-synucleinopathies, cerebrovascular disease) given that each neurodegenerative disease has a different course. While this is acknowledged in some sections, I still don't think it comes through well that different etiologies are associated with different neuropsychological profiles and that those will predict progression, symptom manifestations, etc. As such, I don't think this manuscript adds much to what is already established by the field of neuropsychology.

We have updated the discussion section to explicitly address this point and clarify the limitations regarding generalisability to other dementia etiologies.

2.3 I appreciate the authors' clarification and efforts to emphasize accessibility and scalability. However, while cost and resource constraints are important considerations, neuroimaging remains essential for characterizing the underlying etiologies of MCI and differentiating between Alzheimer's disease, vascular, and other causes of cognitive decline. Given that the ADNI dataset includes rich imaging data, the authors already have access to modalities that could strengthen their models through etiologic subtyping, even if the predictive focus remains on clinical data.

We agree that neuroimaging data provides valuable information for etiological differentiation and disease characterisation. However, the primary aim of our study was to evaluate whether predictive models based solely on routinely collected clinical assessments can provide reliable prognostic insights. The focus reflects a deliberate design choice to maximise accessibility and scalability, particularly in low-resource settings where neuroimaging data is not regularly collected. While the ADNI dataset includes imaging data, incorporating these

modalities would shift the scope of the work away from our core objective of testing the feasibility of clinical assessment-driven prediction. This focus is particularly relevant because, in many real-world contexts, including the NHS, patients often receive an AD or MCI diagnosis without undergoing neuroimaging or CSF collection. Developing models that rely on data that is accessible and available in routine care is essential for creating tools that can be deployed within existing health systems, globally.

2.4 At present, it is not clear that the proposed tool provides clinically actionable insights beyond what is achievable through existing clinical assessment and neuropsychological approaches. The model's utility would be enhanced if it could help improve/predict diagnostic subtypes or imaging markers (which again is something that the field of neuropsychology does). As it stands, the tool's practical contribution to clinical decision-making remains limited.

As we describe in the discussion of the manuscript, the current version of our digital tool has not yet been validated in memory clinics. The study we present in this paper is intended as a proof-of-concept to demonstrate the feasibility of generating individualised predictions of cognitive and functional decline using routinely collected clinical assessment data. A secondary aim of the work was to build a user interface to show how our models could be used in clinical settings.

While predicting neuroimaging and diagnostic subtypes are interesting goals in dementia research, they are not the focus of this study. Our aim was to explore whether accessible, and clinically validated assessments could support more personalised prognostic insights, particularly in settings where advanced biomarkers are not available.

2.5 The authors have now included data from ADNI but they have not described which specific participants from ADNI they have included (e.g., diagnoses, selection process, which ADNI samples) nor have they described patient demographics in the methods.

Thank you for this observation. This may have been unclear in the wording we chose. The ADNI validation cohort's demographics and participant diagnoses were included in the revised Tables 1 and 3 in the Results. We have included more clarification and revised the methods section of the manuscript to describe the selection criteria for Minder and ADNI participants.

2.6 It still remains unclear why the authors believe a data driven approach is superior to traditional statistics

Both traditional statistics and machine learning offer unique and valuable insights (which is why we include both methods in our paper). Conventional statistics typically examine associations between each feature and outcome independently at the population level. While this is valuable for understanding broad trends, it does not capture the complex, non-linear

interactions between multiple features or provide individual predictions. In our study, statistical analyses revealed significant associations between each feature and the outcomes, but they were insufficient for predicting future cognitive or functional scores for individual participants and limited in providing a holistic view of interactions between different factors. Machine learning models can integrate several features simultaneously, model their interactions, and generate patient-specific forecasts, which is critical for personalised care planning. Our aim was not to replace statistical analysis but to complement it by addressing its limitations in predicting individual trajectories. This approach enables us to move from descriptive associations to actionable and personalised prognostic insights, which is the core contribution of this work. We have clarified this distinction in the discussion section.